# Future upstream water consumption and its impact on downstream water availability in the transboundary Indus basin

Wouter J. Smolenaars[1], Sanita Dhaubanjar[2,3], Muhammad K. Jamil[1,4], Arthur Lutz[2], Walter Immerzeel[2], Fulco Ludwig[1], Hester Biemans[1,5]

[1]Water Systems and Global Change Group, Wageningen University, Wageningen, 6708 PB, The Netherlands
[2]Department of Physical Geography, Utrecht University, Utrecht, 3584 CB, The Netherlands
[3]International Centre for Integrated Mountain Development, Kathmandu, 44700, Nepal
[4]Pakistan Agricultural Research Council, Islamabad, 44690, Pakistan
[5]Water and Food Research Group, Wageningen Environmental Research, Wageningen, the Netherlands

*Correspondence to*: Wouter J. Smolenaars (wouter.smolenaars@wur.nl)

**Abstract.** The densely populated plains of the lower Indus basin largely depend on water resources originating in the mountains of the transboundary upper Indus basin. Recent studies have improved our understanding of this upstream-downstream linkage and the impact of climate change. However, water use in the mountainous part of the Indus and its hydropolitical implications have been largely ignored. This study quantifies the comparative impact of upper Indus water usage, through space and time, on downstream water availability under future climate change and socio-economic development. Future water consumption and relative pressure on water resources vary greatly across seasons and between the various sub-basins of the upper Indus. During the dry season, the share of surface water required within the upper Indus is high and increasing, and in some transboundary sub-basins future water requirements exceed the availability during the critical winter months. In turn this drives spatiotemporal hotspots to emerge in the lower Indus where seasonal water availability is reduced by over 25% compared to natural conditions. This plays an important, but previously not accounted for, compounding role in the steep decline of per capita seasonal water availability in the lower Indus in the future, alongside downstream population growth. Increasing consumption in the upper Indus may thus locally lead to water scarcity issues, and increasingly be a driver of downstream water stress during the dry season. Our quantified perspective on the evolving upstream-downstream linkages in the transboundary Indus basin highlights that long-term shared water management here must account for rapid socio-economic change in the upper Indus and anticipate increasing competition between upstream-downstream riparian states.

## 1 Introduction

The Indus basin is shared by Pakistan, India, Afghanistan and China, and is home to over 260 million people(Wada et al., 2019). The basin is among the most depleted and water stressed in the world(Laghari et al., 2012; Wada et al., 2011). The arid plains of the lower Indus basin are densely populated and rely on the largest contiguous irrigation system in the world for their food production. Water demands for irrigation- but also increasingly for domestic and industrial purposes- considerably exceed

the dry season supply of freshwater and are compensated for by the overexploitation of groundwater resources(Karimi et al., 2013; Wijngaard et al., 2018). Despite the current overuse of water resources, progress towards achieving the interlinked food-

, and water security *Sustainable Development Goals* (SDG 2 & 6 respectively) in the Indus basin is insufficient(Rasul, 2014, 2016). Moreover, the direct- and indirect water resources required to meet these SDGs are projected to increase further under pressure from socio-economic development(Smolenaars et al., 2021; Vinca et al., 2020). Achieving and sustaining the food- and water security SDGs in the transboundary Indus basin can only succeed with basin-wide integrated adaptation efforts(Immerzeel & Bierkens, 2012; Immerzeel et al., 2020).

Over 85% of the Indus basin's annual discharge originates from the mountainous and scarcely populated upper Indus (Biemans et al., 2019) , which is shared between all four riparian states. A combination of snowmelt and monsoon rainfall cause mountain water availability across the basin to surge over the Asian summer, while run-off during the dry winter is limited(Laghari et al., 2012). The vast irrigation networks and megacities of the Pakistani and Indian lower Indus plains are therefore highly dependent on the timely provision of mountain water resources(Biemans et al., 2019; Flörke et al., 2018; Wijngaard et al.,

2018), a considerable part of which is transboundary in origin. Previous modelling studies showed that climatic and socio-economic changes may intensify the existing Indus basin upstream-downstream dependencies. Climate change is projected to cause a consistent rise and seasonal shift in upper Indus run-off(Lutz et al., 2014), while population growth, economic progress and urbanization are likely to spur rapid growth of downstream water demands(Biemans et al., 2013; Wijngaard et al., 2018). Consequently, the Indus basin has been framed as containing strong, one-directional upstream-downstream linkages; the

mountainous upper Indus provides and the populous plains of the lower Indus consume water(Khan et al., 2020; Laghari et al., 2012; Reggiani & Rientjes, 2015; Wijngaard et al., 2018). Research investigating the future water resources of the upper Indus basin has accordingly remained largely within the bio-physical domain, exploring the effects of climate change on upstream hydrology and its role as source of water only (Khan et al., 2020; Lutz et al., 2014; Lutz, Immerzeel, et al., 2016; Reggiani & Rientjes, 2015). Regional modelling studies on the influence of anthropogenic activities on the Indus basin water system have

likewise focused on the lower Indus basin(Momblanch et al., 2019; Vinca et al., 2020; Wada et al., 2019; Yang et al., 2016), or simply assumed upstream water use activities to be insignificant(Biemans et al., 2019; Wijngaard et al., 2018). Only Amin et al. (2018) and Mehboob and Kim (2021) explicitly examined the development of water demands in the upper Indus basin. But these studies only covered the upstream parts of the Pakistani share of the basin and did not quantify  downstream or cross-border implications.

However, rapid socio-economic development is not limited only to the lower Indus basin. The upper Indus basin also contains fast emerging urban centres (Kabul, Jalalabad, Peshawar, Srinagar, see figure 1) that will place an increasing claim on water resources in the future(Smolenaars et al., 2021). Upstream anthropogenic activities can exacerbate, or even cause, downstream hydrological droughts(Rangecroft et al., 2019; Van Loon et al., 2016), especially in basins like the Indus where downstream areas rely heavily on water generated by upstream sources(Zhou et al., 2019). Already now, transboundary water allocation

issues in the Indus basin are exacerbating and causing considerable geopolitical tension in the water stressed Kabul sub-basin between upstream areas in Afghanistan and downstream areas in Pakistan(Atef et al., 2019). Global assessments of upstream-

downstream linkages in transboundary basins that quantified future dependencies (Munia et al., 2018; Viviroli et al., 2020) and drivers of water stress (Degefu et al., 2019; Munia et al., 2016; Munia et al., 2020) similarly found the Indus basin at considerable risk for future conflicts. Such studies are however based on coarse approaches that aggregate the basin into

upstream, midstream and downstream units, and provide limited quantitative insight at the level of individual Indus tributaries where transboundary issues, as seen in the Kabul sub-basin, arise in practice. Socio-economic changes in the upper Indus will thus increasingly affect water availability in both the upper- and lower Indus basin and water sharing between riparian states, but the potential magnitude of their influence throughout the basin is presently unclear.

Transboundary water management and adaptation in the context of the SDGs requires a spatially explicit understanding of the
interplay between future water demands and availability, and between upstream and downstream regions (Rangecroft et al., 2019; Yillia, 2016). Additional disaggregated insight into the implications of changing water use activities in the upper Indus on water availability throughout the Indus basin, particularly in relation to climatic changes, is therefore needed. In this study we hypothesize that water consumption in the upper Indus can no longer be ignored, and that it will be an increasingly important driver of transboundary downstream water stress in the coming century. *The aim of this paper is to quantify, both in space and*
*time, the potential impact of upper Indus water consumption on lower Indus water availability accounting for both socio-economic development and climate change.* To do so, validated datasets on Indus hydrology and socio-economic development are combined within a novel water accounting approach that conceptually simulates the complex upstream-downstream dependencies in the transboundary Indus basin. The results provide a first-time quantified perspective on the comparative role of upper Indus socio-economic changes within the broader development of Indus basin upstream-downstream linkages. This
insight is important for long-term shared water management between riparian states, adaptation research and hydrological modelling at the basin and sub-basin scales. The approach presents a novel way forward for regionalised upstream-downstream assessments in other complex transboundary river basins.

## 2 Methods and Materials

### 2.1 Case study description: State of water management in the Indus basin

Since ancient times, the water resources of the Indus river and its tributaries have been used extensively for irrigation practices in the fertile lower Indus plains. The current Indus Basin Irrigation System (or IBIS) was first developed around the 1850's and gradually expanded over many decades to become the largest continuous irrigation system in the world. After Partition in 1947, the IBIS, and the upstream areas that provide it with vital water resources, were divided between India and Pakistan. This major change in riparian relations within the Indus basin led to a highly complex transboundary water management
setting(Zawahri & Michel, 2018). In a bid to improve shared water management the World Bank brokered the Indus Water Treaty (IWT) between India and Pakistan in 1960(Qamar et al., 2019).

The IWT allocates the water resources of the upper Indus between two riparian states (see Figure 1), with Pakistan receiving control over the water of the western tributaries (Indus, Jhelum and Chenab), and India over that of the eastern tributaries

(Ravi, Beas and Satluj). While this allots a majority of Indus water system discharge to Pakistan(Kalair et al., 2019), the three western tributaries originate in- or cross- the Indian share of the basin before feeding into the lower Indus in Pakistan. The IWT therefore allows limited local water use (e.g. irrigation and domestic purposes) and unlimited non-consumptive use (e.g. run-of-river hydropower and transportation) to upstream India in these tributaries(Zawahri & Michel, 2018). Although the IWT has facilitated three notable transboundary water conflicts and regulated hydropolitical relations for more than six decades, many have pointed out the need to update the framework to meet the new challenges imposed by global change(Parvaiz, 2021; Qamar et al., 2019).

The IWT is not the only treaty governing water management and distribution in the Indus basin. In Pakistan, the Indus water system is the sole source of fresh surface water for the large majority of the country. Water allocation between the provinces of Pakistan is consequently arranged via the Pakistan Water Appointment Accord, which distributes available flow roughly by order of water demand over the four Pakistani provinces(Basharat, 2019). This framework has been shown to work well in high-flow periods, but intra-national disputes have occurred in years of drought, with downstream regions claiming to receive consistently less water than what should be allotted to them(Hassan et al., 2019). Other regions of the Indus basin are not governed by transboundary treaties. The most prominent of these is the Kabul river basin, one of the largest tributaries of the Indus river and a major source of fresh water for both Pakistan and Afghanistan(Qamar et al., 2019). Similarly, upstream China is not part of any water sharing agreement in the Indus basin, but its claim on water resources has so far remained limited due to the low population density and mountainous terrain of its share of the basin(Zawahri & Michel, 2018).

In this study, we used the context of the IWT and shared water management in the Indus basin, as described here, to shape our water accounting approach- both in terms of spatial resolution and in the water use sectors that we consider. In addition, we reflect on the implications that our results may hold for this shared water management context in the discussion section.

## 2.2 Upstream-downstream water accounting approach

To quantify the impact of upper Indus water usage on downstream water availability we used a water accounting approach at the sub-basin level of individual Indus tributaries, and at seasonal timescale. We applied this approach to assess future changes for two integrated climatic and socio-economic change scenarios over the period 1980-2080. For both scenarios, our approach consisted of two assessment steps. First, we quantified the development of upper Indus water availability under climate change and subtracted future water consumption. Then, we allocated remaining upstream water over downstream sub-basins and assessed downstream water availability, with and without accounting for upstream consumption. The distribution of remaining water from upstream sub-basins over their respective downstream sub-basins was determined using a novel upstream-to-downstream allocation algorithm developed in this study (see Figure 2 and Section 2.5.3). Water availability in our approach is operationalised as the per capita available water resources in $m^3yr^{-1}$, as this accounts for the effect of population changes on the relative water resources available for socio-economic activities (Hanasaki et al., 2018). In the following sections we explain in more detail the spatiotemporal resolution and methods that comprise our approach, and the scenarios and data we used to apply it for our Indus basin assessment.

### 2.3 Spatial and temporal disaggregation

### 2.3.1 Sub-basin delineation

Previous studies that quantified transboundary upstream-downstream linkages(Degefu et al., 2019; Munia et al., 2016; Munia
et al., 2018; Munia et al., 2020),  used approaches that divide river basins into two or three sub-basins with a linear flow of
water between them. Similarly, our study was also conducted at the sub-basin level. However, instead of assessing the entire
upper Indus as one lumped sub-basin, our approach defined sub-basins for each of the main tributaries subject to the IWT (see
Figure 1 and Section 2.1), and the Kabul river. Sub-basins (see Figure 1) were delineated using a pour point analysis in ESRI
ArcGIS with a 5 arcmin drainage direction map from Hydrosheds (Lehner et al., 2006). First, the upper Indus sub-basins were
established by determining the upstream area of the Indus river and its main tributaries. For each river course, the cut-off
between upstream and downstream was set at major dams situated within the mountain-to-plain transition zone, which is an
often used definition in Indus basin hydrology(Lutz et al., 2014; Lutz, Immerzeel, et al., 2016; Wijngaard et al., 2018). The
contributing area upstream from these locations were assessed and resulted in seven sub-basins that were named after their
respective main river (see Figure 1).
To facilitate the spatially explicit assessment of downstream impacts due to upper Indus water use, the connectivity between
the lower Indus and the upper Indus sub-basins needed to be established. Similar to our upstream delineation of sub-basins,
we disaggregated the lower Indus basin into multiple sub-basins, based on the overlapping downstream areas of upper Indus
sub-basins. Specifically, we delineated lower Indus sub-basins  at the confluences of rivers originating from the upper Indus
basin. These sub-basins are thus defined by the upper Indus tributaries they receive water from. This allowed our approach to
assess which areas within the lower Indus are particularly affected by upstream consumption, whereas beforementioned
lumped approaches only provided insight into the upstream-downstream linkage of the basin at large. The distribution of
mountain water throughout the lower Indus basin is however highly controlled by an expansive system of barrages and linkage
channels(Wescoat Jr et al., 2018). This infrastructure plays a key role in Indus basin water management as it allows riparian
states to optimally distribute their scarce water resources(Basharat, 2019). The water flows through the most important linking
canals (Indus-Jhelum, Jhelum-Chenab-Ravi-Satluj and Chenab-Ravi, see figure 1) were therefore also considered in the
delineation of downstream sub-basin and the designation of the downstream area of upper Indus sub-basins. This approach
resulted in eighteen lower Indus sub-basins that each receive water resources from a unique combination of upper Indus sub-
basins (see Figure 1).

### 2.3.2 Seasonality and timeframe

The strong seasonal character of Indus hydrology requires water resource assessments to be conducted at the seasonal
level(Laghari et al., 2012). Therefore, contrary to the annual level of previous studies(Munia et al., 2016; Munia et al., 2018;
Munia et al., 2020; Viviroli et al., 2020) we aggregated and analysed hydrological changes and impacts for the two hydrological
seasons suggested by Laghari et al. (2012), that correspond with the main agricultural season; the *Dry season* (Rabi cropping

season, Nov-Apr) and the *Wet season* (Kharif cropping season, May-Oct). Additionally, for some analyses the seasons were disaggregated further to the four climatological seasons used in other regional water system studies(Rajbhandari et al., 2015; Wijngaard et al., 2018); *Pre-monsoon* (Mar-May), *Monsoon* (Jun-Aug), *Post-monsoon* (Sep-Nov) and *Winter* (Dec-Feb). To illustrate the progression of water consumption and availability over time, data was assessed as transient annual timeseries or for three assessment timesteps; the 1980-2010 historical reference period (Ref), and the future 2030-2050 (Mid) and 2060-2080 (Late) periods.

## 2.4 Integrated scenarios

Both climate and socio-economic change might increase pressure on available water resources. To obtain insight into potential future changes in upstream-downstream linkages and impacts, we defined two regional scenarios that integrate socio-economic development and climate change. The socio-economic core of the scenarios was sourced from a set of regionalised and spatially downscaled *Shared Socio-economic Pathways* ('SSPs', see O'Neill et al. (2014)) specifically downscaled towards 2080 for the Indus basin by Smolenaars et al. (2021). The optimistic 'SSP1-Prosperous' (sustainable economic progress and low population growth, hereafter: SSP1) and the pessimistic 'SSP3-Downhill' (fragmented economic stagnation and high population growth, hereafter: SSP3) storylines were selected, as these provided the highest contrast and thus the broadest plausible bandwidth of results.

The socio-economic storylines are regionally downscaled extensions of the global SSP storylines and could therefore be consistently matched with the RCP emissions framework(van Vuuren et al., 2014). To represent future climatic conditions we combined the SSP1 and SSP3 storylines with respectively the moderate RCP4.5 and extreme RCP8.5 emission scenarios. This resulted in two future scenarios for climate, population and GDP: SSP1-RCP4.5 and SSP3-RCP8.5 (hereafter referred to as SSP1 and SSP3).

## 2.5 Upstream-downstream assessment and data sources

## 2.4.2 Scenario forcing data

Applying the two integrated scenarios within our quantitative upstream-downstream approach required us to obtain spatially explicit climatic and socio-economic forcing data for our scenarios (see Table 1). For the socio-economic storylines, spatially explicit future population projections towards 2080 at 5 arcmin (~8 km) resolution are available that account for population growth and urbanisation, as well as downscaled GDP projections(Smolenaars et al., 2021). For the 1980-2010 reference period, we used the 5 arcmin population maps of HYDE project (Klein Goldewijk et al., 2011). Historical GDP data was obtained from IIASA(Dellink et al., 2017). Climate change projections at daily timescale for the coupled RCP emission scenarios were obtained from eight (four per RCP) downscaled GCM projections for the wider South Asia region at 5 arcmin resolution over the period 1980-2100 (Lutz, ter Maat, et al., 2016).

**2.4.3 Determining the impact of upper Indus water consumption on remaining water availability**

As the first assessment step of our approach, we determined for both scenarios the progression of water consumption in the upper Indus basin in relation to the change in water availability under socio-economic development and climate change. For the upper Indus sub-basins, daily natural discharges were determined at the sub-basin outlets (i.e. the absolute surface water availability per sub-basin). Validated high-resolution discharge projections for the seven upper Indus sub-basins were used at daily timesteps for the reference period and for both RCPs (1980-2100) (Wijngaard et al., 2018; Wijngaard et al., 2017). These

projections are generated by the distributed *Spatial Processes in Hydrology* (SPHY) cryosphere–hydrology model based on the same downscaled climate forcing data that pertains to the climatic scenarios of this study. The SPHY model was developed specifically to simulate the glacier-dominated hydrology of High Mountain Asia and has been often been applied for the Indus basin(Biemans et al., 2019; Lutz et al., 2014; Lutz et al., 2019).

Subsequently, we decreased the daily natural discharges with daily aggregated consumptive water requirements for the

domestic, industrial and agricultural sector of each sub-basin to estimate actual discharge. Consumptive water requirements were defined as the sectoral water demands, minus the return flows(Bijl et al., 2016), which represent the amount of natural water resources that are made unavailable for downstream usage. Consumptive water requirements in excess of daily surface water availability were assumed to be stored within the sub-basin in the closest preceding days with surplus discharge and released on the day shortages occurred. The difference between natural and actual outflow of upper Indus sub-basins therefore

always equalled the consumptive requirements at the annual level, but for daily timesteps these occasionally varied. Sectoral consumption data were obtained from the following sources:

- Domestic and industrial consumptive water requirements projections for the upper Indus basin were obtained with the regression models of Bijl et al. (2016). The models simulate annual water consumption intensity per sectoral unit (capita and $US of GDP respectively) as a product of economic development (expressed in GDP per capita) increasing

efficiency through time, and a pre-calibrated 'region-factor' that accounts for climatological and cultural circumstances (see Appendix 1). The models were forced for each basin-country with the national-level GDP per capita projections of the scenario forcing data. As the Bijl models provided an annual consumption value, daily consumptions were assumed to be 1/365[th] of the annual output and thus to not vary within the projected year. The simulated daily consumption intensities were multiplied by the projected total population and GDP of the basin-share

of each country, and then spatially distributed over the gridded population projections of the scenarios. Population data for both the reference and projected periods was available at 10 year timesteps in the forcing dataset. To obtain annual values the data was linearly interpolated between these timesteps. Lastly, the gridded consumption data was summed for each upper Indus sub-basin.

- To obtain water usage data for the agricultural sector the grid-based integrated crop production-hydrology *Lund–*

*Potsdam–Jena managed Land* (LPJmL) model was used. LPJmL simulates water balance and crop production for twelve crops (irrigated and rainfed), and the interaction between them, whilst considering for climatic circumstances

and anthropogenic interventions(Bondeau et al., 2007). This allows the influence of crop production on the water system to be quantitatively untangled and studied under climatic and socio-economic changes(Gerten et al., 2011; Rost et al., 2008). For this study a regional LPJmL version was used that was developed specifically to represent the monsoon-dominated double-cropping systems of South Asia at 5 arcmin resolution (see Biemans et al. (2019)). The South-Asia LPJmL version has been applied for multiple integrated assessment that include the Indus basin(Biemans et al., 2019; Wijngaard et al., 2018) and its agricultural water withdrawals have been validated for the broader South Asia region(Biemans et al., 2016; Biemans et al., 2013). The LPJmL simulations were conducted with unlimited groundwater access for irrigation, providing an estimate of the potential agricultural water consumption. This avoids inconsistencies with the discharge data obtained from the SPHY model. LPJmL was forced with the downscaled climate data pertaining to the scenario dataset and with regional land-use based on land-use change projections for SSP1 and SSP3 from the IMAGE integrated assessment model (Stehfest et al., 2014). The land-use projections were constructed at 5 arcmin resolution by applying the IMAGE growth-rates for rainfed and irrigated crops to 2005 land-use extents from the spatially explicit MIRCA-2000 dataset(Portmann et al., 2010), an approach that is often used for scenario based  studies with LPJmL(Wijngaard et al., 2018). The daily consumptive water requirements were determined by aggregating the blue water consumption (i.e. evapotranspiration originating from blue water (surface and groundwater) resources) of agriculture from evapotranspiration and conveyance losses and summing these per sub-basin. Surface water in LPJmL is only extracted if there is a soil moisture deficit. This agricultural green water footprint (i.e. evapotranspiration originating from green water (precipitation) resources), was not considered in the total agricultural water usage, as the SPHY discharge projections already account for green water evapotranspiration through a natural vegetation layer (Wijngaard et al., 2017).

To further interpret the consequences of climatic- and socio-economic changes on the status of water availability in the upper Indus basin the APC (Availability Per Capita) index(Hanasaki et al., 2018) was applied, which is an expanded version of the well-known Falkenmark index(Falkenmark et al., 1989). The APC index assesses the annual available water resources per capita and categorises these by the degree to which water scarcity is limiting a society:

- No water stress: >5000 $m^3$ per capita per year
- Low water stress: 5000-1700 $m^3$ per capita per year
- Moderate water stress: 1700-1000 $m^3$ per capita per year
- High water stress: 1000-500 $m^3$ per capita per year
- Extreme water stress: <500 $m^3$ per capita per year

Lastly, the impact of upper Indus consumption on environmental flows was studied using the *variable monthly flow* (VMF) method as applied by Pastor et al. (2019). VMF defines that a minimum of respectively 30% and 60% of mean natural flows in the dry and wet seasons must be maintained for environmental well-being. Thus, only 70% and 40% of water resources during the wet- and dry season can sustainably be consumed(Pastor et al., 2014). Minimum daily flow thresholds were determined for the mean daily flows over the historical reference period (1980-2010) and the wet and dry season definition by

Laghari et al. (2012). The status of environmental flows was expressed as the days per year in which minimum flows are not met at the outlet of upper Indus sub-basins.

### 2.4.4 Quantifying upstream-downstream linkages and impacts

For the second assessment step of our approach we quantified the impact of upper Indus consumption on water availability in
the lower Indus. This step required surplus water resources in upper Indus sub-basins to be allocated over the lower Indus sub-basins. Previous studies (Degefu et al., 2019; Munia et al., 2016; Munia et al., 2018; Munia et al., 2020) used a linear method for this upstream-to-downstream water allocation, meaning that surplus water flows from an upstream sub-basin to one fixed downstream sub-basin. However, our water accounting approach considered for multiple upstream sub-basins, with an overlapping mesh of downstream sub-basins. We moreover accounted for linkage channels (see section 2.3.1) when defining
the downstream area of each upper Indus sub-basin. This means that the downstream distribution of surplus upstream water is not only based on natural flow direction, but is also demand based and thereby inherently variable. Beforementioned linear methods were thus not suitable to simulate upstream-to-downstream water allocation in our regionalised approach.

We therefore developed a new routine (see Figure 2) that works similar to the approach of Viviroli et al. (2020), which distributes surplus upstream water resources equally over all downstream grid cells. Instead of distributing surplus upstream
water on the basis of geographical area, however, we distributed it based on population, as we think this is a better proxy for where water demand is located. Our upstream-downstream water allocation algorithm assumes an equitable distribution of upper Indus outflows among the downstream population of each upper Indus sub-basin. Populations of lower Indus sub-basins that are downstream from multiple upper Indus sub-basins were divided and assigned to upstream sub-basins relative to the water supplied (see Figure 2). This allowed for the simultaneously allocation of upstream-downstream water resources for all
upper Indus sub-basins, without having to make quantitative assumptions as to how water is distributed between multiple competing downstream sub-basins.

We applied this upstream-to-downstream allocation routine for the three assessment timesteps (Ref, Mid and Late). First, the average natural flow and average actual flow were determined per season and then distributed over the lower Indus sub-basins. The allocation procedure used the spatially explicit population projections of the scenario forcing data set as population input
data for lower Indus sub-basins. The total water availability of each lower Indus sub-basin was then determined by aggregating, for each timestep and season, the allocated upper Indus water resources with average water supply generated within the lower Indus sub-basin itself. Hereby, it was assumed that all water resources generated in a lower Indus sub-basin are utilized within that sub-basin. The water resources originating locally in the lower Indus sub-basins were determined with the LPJmL model. Simulations were ran with naturalized upstream inflow, natural vegetation and without anthropogenic water withdrawals, an
approach that is often used to determine natural flows (Jägermeyr et al., 2017; Rost et al., 2008). The model was forced with the downscaled climate data of the respective scenarios. For each of the lower Indus sub-basins, the discharges at its outlet were assessed and the inflows from outside the sub-basins were extracted (i.e. the discharges at the outlets of sub-basins directly feeding into a sub-basin), thus leaving only the water generated within the sub-basin itself.

The impact of upper Indus consumption on lower Indus water availability was then studied by comparing relative differences in total seasonal water availability between the actual and natural flow conditions for each timestep. As availability between seasons and sub-basins varied greatly, the absolute and annual based APC index was not suitable for this analysis. Water availability in the future timesteps was additionally compared to reference period availability to assess the change in lower Indus water availability through time under integrated climate change and socio-economic development. This provided insight into the comparative role of upper Indus water consumption. Similarly, per capita water availability in the lower Indus in our approach was also affected by population growth, and by climate change through its effect on discharges. We therefore additionally assessed water availability in lower Indus sub-basins was for future timesteps with downstream population distributions and climatic conditions independently kept in reference period conditions (i.e. with population maps and discharges as they were in the Ref 1980-2010 timestep). This allowed the isolated effects of respectively climate change and downstream population changes on future water availability in the lower Indus to also be quantified and compared to the impact of upper Indus consumption.

## 3 Results

### 3.1 Changes in upper Indus water consumption

Figure 3B shows that the reference period total water consumption in the upper Indus basin is around 6.9 km$^3$ yr$^{-1}$ (compared to approximately 140 km$^3$ yr$^{-1}$ in the lower Indus basin (Wijngaard et al., 2018)) Water use activities are mostly located in the Kabul, Indus and Jhelum sub-basins and are dominated by agricultural water use during the wet season. The population in the upper Indus is projected to grow by 124% and 245% towards 2080 in SSP1 and SSP3 respectively (Table 2, compared to reference period 1980-2010). The highest population growth will be in the Kabul sub-basin (188% in SSP1 and 350% in SSP3), especially in the Afghani share (Smolenaars et al., 2021). This sub-basin contains three large cities, two of which in Afghanistan, that are projected to expand rapidly due to the strong urbanization trends (see Figure 3A). Water consumption in the upper Indus subsequently demonstrates an annual growth to 13 km$^3$ yr$^{-1}$ (88%, SSP1) and 17 km$^3$ yr$^{-1}$ (146%, SSP3) in the 2060-2080 period. Consumption increases are largely concentrated in sub-basins that already account for the majority of present water usage. The Kabul and Jhelum sub-basins are projected to face annual water use increases of respectively as much as 135% and 307% in the SSP3 late period, with this growth largely located in the respective Afghani and Indian parts.

The projected growth in water consumption is highest for the domestic sector (figure 3B). Population growth and economic progress are projected to increase both the number of end-users and the amount of consumed water resources per end-user. Economic growth similarly drives an increase in the industrial water use. Agricultural water use only increases slightly from present day values as expansion options in the mountainous upper Indus are limited and higher temperatures due to climate change reduce the length of the growing season of staple crops (Wijngaard et al., 2018). The relative growth in the domestic- and industrial water use-dominated dry season (179% in SSP1 and 296% in SSP3) is therefore greater than in the wet season

(60% in SSP1 and 102% in SSP3) and the annual average (see Appendix 4). Figure 3 shows that the seasonal difference in water consumption in the upper Indus basin is accordingly projected to decrease by the late period in both scenarios.

**3.2 Impact of climatic and socio-economic changes on upper Indus water resources**

Table 2 demonstrates that the ensemble mean annual flow of the upper Indus increases by 38% and 32% respectively in the SSP1 and SSP3 scenarios for the 2060-2080 period. The heightened discharge is consistent between the two scenarios, as both
predict temperatures in South Asia to increase (~2°C in RCP4.5 and ~5°C in RCP8.5, see (Lutz, ter Maat, et al., 2016)), which drives increased glacial melting until at least the end of the century (Wijngaard et al., 2017). The relative increase is most pronounced in the dry season. The development of discharge does nonetheless vary greatly between the sub-basins. The Satluj and Indus sub-basins are projected to face annual flow increases of up to 54% and 51% respectively, while those of the Kabul and Jhelum sub-basins stay roughly similar over the projected period.

Despite the general increase in surface water availability, the mean annual per capita water availability in the upper Indus basin is projected to drop by 43% (SSP1) and 65% (SSP3) by the late period under pressure from rapid population growth (Table 2). The application of the APC index in Table 2 illustrates that the upper Indus basin as a whole is projected to drop from a 'no water stress' situation in the refence period to a 'low water stress' situation in the mid period of both scenarios. However, the per capita water availability change is highly heterogenous between the sub-basins. In the reference period the relatively
densely populated and transboundary Kabul and Jhelum sub-basins fall into the 'low water stress' category of the APC index and are projected to move into the 'high-' and 'moderate' water stress categories in the late period of the SSP3 scenario, largely due to rapid population growth surrounding major urban centres in the Afghani and Indian shares of the respective basins (Smolenaars et al., 2021). In contrast, other sub-basins, such as Satluj, Chenab and Ravi, all located largely in India, remain firmly in the 'no water stress' category and even face a net increase in per capita water availability in the SSP1 scenario due
to the positive effect of climate change on discharges here.

Figure 4B demonstrates that during the refence period the consumed share of total annual surface water resources is negligable at about 2%. Because of the seasonal discharge patterns the consumption in the driest (winter) period of the year does exceed 10% of total discharge (Figure 4A). Despite rapid population growth the share of total annual water resources consumed in the upper Indus basin only increases to 4.1% and 5.5% in SSP1 and SSP3 respectively in the late period (see Appendix 2).
However, the basin-level consumed fraction on average reaches a considerable 15% (SSP1) and 18% (SSP3) over the entire dry season and exceeds 30% during the December and January months. Corresponding to the pace of population growth, the development of relative water consumption differs between sub-basins. In the Kabul sub-basin consumptive needs during the late period in the driest months of the year exceed 80% of available surface water on average and even fully surpass it in low discharge years. In the SSP3 scenarios the consumed share during the wet season also reaches a considerable 17% to 21%
(SSP1 and SSP3 respecitvely). Similarly, in the Jhelum sub-basin the average consumed share over the entire dry season reaches 18% (SSP1) and 23% (SSP3) in the late period and consumptive needs during the winter months may exceed

discharges in the driest years. Sub-basins with positive discharge changes due to climate change and low population growth, such as Satluj, remain virtually unaffected in both scenarios.

The rapid increase in consumptive water needs relative to water availability during the dry season is projected to affect environmental flows in the Kabul and Jhelum upper Indus sub-basins. Figure 5 illustrates that in these basins by 2080 environmental flows are on average not met for roughly half- (Kabul) and a third of the year (Jhelum). Environmental flows appear to also gradually be affected in the Chenab and Beas sub-basins during low discharge years. On the other hand, environmental flows in the Satluj and main Indus sub-basins see very limited impact in the present and will remain largely unaffected over the course of the the century. In some scenarios and timestep, the impact even decreases compared to the present. This is especially true in the Satluj sub-basin, where the increase in flow due to climate change is far larger than the increase in water consumption due to socio-economic changes (see Table 2). Environmental flows are least affected during the monsoon season .

### 3.3 Future downstream water availability under socio-economic- and climate change

The influence of upper-Indus consumption on the per capita water availability in the lower Indus basin (see Appendix 5) varies greatly between the seasons. Analogous to the periods of the year in which the consumed share in the upper Indus is highest, Figure 6 illustrates that its impact on downstream water availability is most pronounced in the winter season. During the reference period some sub-basins in the Pakistani Punjab are already shown to be slightly affected in the order of 8% to 12%, but in the late period the available water here may reduce by more than a quarter on average. However, the impact during the post-monsoon season demonstrates the most considerable rise. Several Pakistani sub-basins shift from being largely unaffected during the reference period to facing mean water availability reductions of 14% (SSP1) and 20% (SSP3) in the late period. The influence on water availability during the monsoon season doubles in most basins, but nevertheless does not exceed 6%. Throughout all seasons the impact of upper Indus consumption is strongest in the sub-basins that receive their water from the Kabul and Jhelum upper Indus sub-basins. Additionally, sub-basins with limited local per capita water availability (e.g. due to high population densities or extremely arid conditions) will be more affected, as their relative dependency on mountain water resources is higher. The regional urbanization trend and subsequent spatial concentration of population magnifies this effect in several sub-basins containing large cities. The pattern of basins most affected by upstream consumption is similar between the scenarios, but the degree of impact is higher in the SSP3 scenario.

The impact of upper Indus consumption on lower Indus water availability is not an isolated process, but intertwined with climate changes and with socio-economic changes in the lower Indus itself. Table 2 and Figure 4B demonstrated that climate change causes an increase in discharge from the upper Indus basin and for the lower Indus a slight increase in precipitation is also projected(Lutz et al., 2019). The isolated impact of climate change (Figure 7) likewise increases late period per capita water availability in most lower Indus sub-basins by 20% to 50% compared to reference period climatic conditions. In the areas downstream from the Beas and Satluj upstream sub-basins, largely located in the Indian Punjab and Haryana states,  this increase may even exceed 50%. The increase in downstream water availability from climate change outweighs the decrease

due to upper Indus water use, except in the sub-basins in Pakistan that are directly downstream from the Kabul and Jhelum sub-basins during the dry season in SSP1. Figure 7 moreover demonstrates that lower Indus population growth from an average of 168 million inhabitants over the reference period to 267 million in the SSP1 late period (see Appendix 3) cause a 20% to 50% decrease in per capita water availability of most sub-basins. Rapid population growth to 443 million inhabitants in the SSP3 scenario drives an almost universal decrease of over 60%.

Accordingly, the combined impact of climate change and socio-economic development in the upper Indus largely results in a net increase in the absolute water available to lower Indus sub-basins. However, population growth in the lower Indus basin also requires these resources to be shared among more recipients. The absolute dependency of the lower Indus basin on water resources originating in the upper Indus basin thereby increases. The integrated effect of these processes drives the mean per capita water availability for the majority of lower Indus sub-basins in the SSP1 late period to reduce by 10% to 40% compared

to reference period availability, with only the sub-basins in the Indian share of the basin, downstream from the Beas sub-basin, showing slight increase (see Figure 7). In SSP3 the integrated drivers cause a general reduction between 40% to 60%. The double sided negative effects of socio-economic development on lower Indus water availability thus outpace the positive effect of climate change.

## 4 Discussion

### 4.1 Limitations and opportunities for future research

This study quantified the development of water consumption in the upper Indus basin and its effect on water availability in the lower Indus basin. The water accounting approach that was applied to obtain these results by design is a simplified conceptual representation of the complex Indus basin water system, as this allowed the broader patterns of upstream-downstream dependencies to be assessed. The methodological approach influenced the quantifications presented in this study and their

implications.

Primarily, upper Indus consumption was assumed to be fulfilled exclusively with surface water resources generated seasonally within the sub-basins. In reality, there may be spatial mismatches or quality related preference that cause part of upper Indus water demands to be fulfilled by unsustainable groundwater extractions. Groundwater reservoirs may moreover perform a modulating role between seasons, with excess surface water resources infiltrating in wet periods to be used in times when

water is scarce. Around the city of Kabul groundwater levels have however dropped considerably over the last decades(Mack et al., 2013). Similarly, on the lower Indus plains, groundwater resources are an important supplementary source for urban and agricultural water demand(Basharat et al., 2015; Biemans et al., 2019; Wijngaard et al., 2018).But these resources are also depleting rapidly, especially in the Indian Punjab (Richey et al., 2015; Salam et al., 2020). The impact of upper Indus basin water consumption on water availability in the lower Indus in the dry season will remain subdued while these resources are

still available. This does however imply that groundwater dependency, and thereby overextractions, are likely to aggravate. Due to a lack of spatial coverage in observational data, the availability and long-term durability of groundwater resources in

the upper Indus basin remain uncertain(Cheema et al., 2014; Qureshi et al., 2010; Salam et al., 2020). More research into the status and development of groundwater here is required so that it may be considered in future research steps.

Water quality issues can similarly play an important role in upstream-downstream relations(Wolf, 2007), as exemplified by transboundary water quality disputes emerging in the Chenab and Jhelum sub-basins(Ahmad & Iqbal, 2016; Zawahri & Michel, 2018). Return flows from domestic, industrial and agricultural water usage upstream may be polluted and reduce the downstream availability of water that is of usable quality(Yoon et al., 2015). However, water stress and availability in our analysis are operationalized using indicators for water quantity and do not consider the impact of reduced water quality. The water stress experienced in the lower Indus due to expanding upstream activities may hence be higher than the reduction in availability projected in this study, if pollution prevention measures are not taken. Follow-up research could expand the water accounting analysis applied in this research with water quality indicators for a more holistic assessment of future upstream-downstream linkages. Such analysis may additionally reflect on increasing pollutions with socio-economic development and the need for pollution prevention measures to curb water stress.

In our upstream-to-downstream allocation routine, we moreover assumed upstream outflows to be distributed equitably over all downstream inhabitants. Water use activities in the lower Indus sub-basins were thereby not considered. However, inhabitants closer to upper Indus sub-basins may consume more upstream water than their allocated share and reduce water availability further downstream. Other lower Indus sub-basins with surplus local water resources may positively affect water availability in other sub-basins. On the other hand, intra-national water sharing treaties, such as the Pakistani Water Appointment Accord, do ensure that upstream water distribution throughout the lower Indus basin is not determined solely on the independent self-interest of each downstream region(Hassan et al., 2019). The results of this study thus provide quantified insight into general trend of lower Indus water availability and the times and areas most likely to be affected by changing upper Indus water use activities from an intrinsic upstream-to-downstream perspective, instead of fully disaggregated quantifications of future water distribution in the lower Indus basin.

High-resolution spatial information on the development of water resources is however required to support data-driven water management and adaptation policy making to support the SDGs(Laghari et al., 2012; Rangecroft et al., 2019; Yillia, 2016). Our assessment made considerable gains in this regards compared to previous upstream-downstream studies, but further spatial disaggregation with fully distributed models and the subsequent inclusion of adaptation measures to curb water stress are important follow-up steps for robust adaptation planning. Accounting for the unique regional, often socio-economic, characteristics that govern water distribution in transboundary rivers basins is challenging in data-intensive and process-based hydrological models. In this light, our conceptual approach offers a valuable alternative to establish initial benchmarks. Our accounting routine provides disaggregated insight into potential hotspot areas and seasons for upstream-downstream impacts and its drivers, with only limited data requirements and a flexible and transparent water allocation mechanism. This approach could similarly be applied to study future upstream-downstream linkages in other complex transboundary basins such as the Mekong and the Nile(Johnston & Smakhtin, 2014). Follow-up research could additionally perform a similar assessment to quantify hydrological interactions between sub-basins within the lower Indus. The relation between the irrigation-dominated

plains of the Indus midstream and the hyper-arid delta could be of particular interest(Laghari et al., 2012). Similarly, more insight into the interplay between socio-economic and climatic drivers for future upstream-downstream linkages in the Indus basin is important, for example by using different, less conventional, RCP-SSP scenario combinations.

**4.2 Implications for future transboundary water management and adaptation planning**

The quantifications presented here provide valuable initial insight into the increasing relevance of water use activities in the upper Indus for the basin's upstream-downstream linkages and hydro-politics. Consistent with other studies(Vinca et al., 2020; Viviroli et al., 2020; Wijngaard et al., 2018), per capita water availability in the lower Indus was shown to decrease over the projected period under integrated climatic and socio-economic changes, while the dependency on upstream water resources increases. Within this development, the reduction in average annual lower Indus water availability, that can be contributed to

expanding water consumption in the upper Indus, remains limited between 4% and 5%. This is in a similar range to study outcomes by Munia et al. (2016) and Degefu et al. (2019), who found current upper Indus consumption to increase downstream water stress by respectively 2% to 4% and 1% to 5%.

However, our results also demonstrate that, when using a spatio-temporally disaggregated approach, hotspot seasons and sub-basins emerge in the lower Indus where the reduction in water availability due to upstream consumption can exceed 25%.

Most affected hereby are the densely populated and rapidly urbanizing central Indus plains of Pakistan, downstream of the Jhelum and Kabul sub-basins, during the dry winter season. The upstream areas and water use activities of these sub-basins are located in the Afghani and Indian shares of the basin respectively. The disaggregation of water availability drivers additionally demonstrated that these upstream changes compound a larger decrease in downstream per capita water availability due to population growth, especially in sub-basins with major cities. This suggests that growing upstream consumption will

considerably contribute to increasing transboundary water stress in the lower Indus in the dry period of the year in which pressure on water resources is already highest (Wijngaard et al., 2018). Systemic adaptive changes to the irrigation-dominated lower Indus water system, as proposed by previous studies (Immerzeel & Bierkens, 2012; Immerzeel et al., 2020; Vinca et al., 2020; Wada et al., 2019), are thus needed to ensure long-term downstream water security here. Our study highlights however that these efforts, and modelling studies in support of them, must explicitly account for changing upper Indus water use and

its implications for water availability downstream.

This study furthermore provides novel insight into the future water balance of upper Indus sub-basins. Strong population growth around the largest urban centres of the upper Indus was demonstrated to cause the Jhelum and Kabul sub-basins to become water stressed themselves by the second half of the century. During the low-flow winter season consumptive water requirements here will consistently claim the majority of available surface water. The actual water demands required to satisfy

consumptive requirements are manifold higher(Bijl et al., 2016) and can likely structurally not be met. This indicates that adaptive changes to regional water management and water use behaviour are essential to mitigate water scarcity issues and achieve water security SDGs, not only in downstream Pakistan, but in the Indian and Afghani shares of these upstream sub-basins as well. During the wettest period of the year over 90% of surface water remains available. A valuable adaptation avenue

suggested by Amin et al. (2018) may therefore lay with modulating seasonal difference with storage dams specifically for upper Indus water provision.

However, the Kabul and Jhelum are transboundary sub-basins. Past plans to construct additional hydropower dams, with limited storage capacity, in the Indian share of the Chenab sub-basin have led to disputes over fears that this infrastructure could be used to further control the flow of vital dry season water resources to downstream Pakistan and infringe on the terms of the Indus Water Treaty(Ahmad & Iqbal, 2016). Both the increasing upstream water use projected for these sub-basins, and hydrological interventions to facilitate this use such as storage dams and diversion canals, may therefore intensify upstream-downstream water competition and aggravate existing hydro-political tensions between the riparian states(Atef et al., 2019; Gupta & Ebrahim, 2017). Transboundary water competition may further exacerbate as downstream demands in the heavily irrigated and densely populated Pakistani and Indian Punjab are also expected to increase with substantial projected population growth, particularly in the SSP3-RCP8.5 scenario (Wijngaard et al., 2018). This demand is most likely to be met with increased use of upstream water and may prompt riparian states to capitalize to even greater extent on upper Indus water resources allotted to them in the Indus Water Treaty(Zawahri & Michel, 2018).

The results of this study therefore support the claims of previous studies that the Indus Water Treaty needs to be revisited(Ahmad & Iqbal, 2016; Kalair et al., 2019; Qamar et al., 2019; Wada et al., 2019) and include the Kabul tributary, and thereby Afghanistan (Zawahri & Michel, 2018), to ensure equitable and sustainable future water allocation between riparian states and provide a robust platform for the development of basin-wide adaptation strategies. The role of climatic changes in this process has been at the forefront of scientific attention(Kalair et al., 2019; Qamar et al., 2019) and policy making(Parvaiz, 2021) in recent years. However, our quantifications show that socio-economic changes may have a larger influence on future upstream-downstream linkages in the basin and the subsequent water stress experienced by its inhabitants. This suggests that any revisitation of existing treaties, like the IWT, towards improved shared water management must account for future socio-economic changes in both the upper and lower Indus basin, alongside the role of climatic change. We specifically identified several transboundary interactions that are likely to intensify in the future and must be addressed accordingly in this process. These hotspots moreover provide targets of special consideration for transboundary cooperation, adaptation policy making and future hydrological modelling studies in support of the integrated pursuit of water and food security SDGs.

## 5 Conclusion

This study quantified the role of current and future water use in the upper Indus on downstream water availability for two integrated socio-economic development and climate change scenarios. The results demonstrate that growing water usage in the upper Indus basin is a significant factor in the evolving upstream-downstream linkages of the Indus basin. The combined consumption across the seven upper Indus sub-basins is projected to increase from 6.9 km$^3$ yr$^{-1}$ presently to 13-17km$^3$ yr$^{-1}$ by 2060-2080. This causes considerable pressure on surface water resources in the dry season. The transboundary Kabul sub-

basin, shared by Afghanistan and Pakistan, and the Jhelum sub-basin, shared by India and Pakistan, in particular are demonstrated to become increasingly water stressed due to rapid population growth, despite an increase in surface water availability through climate change. Water requirements during the critical winter months here may structurally exceed 50% (Jhelum) and 90% (Kabul) of surface water availability in the future and increasingly impede environmental flows from being met. Scarcely populated upstream sub-basins, such as Satluj and Ravi in the Indian share of the basin, instead see the effects of climate change come out ahead and face an overall increase in future water availability.

The large differences in relative upper Indus water consumption between seasons and sub-basins result in spatiotemporal impact hotspots in the lower Indus where surface water availability is reduced by over 25% compared to natural flow conditions. This amplifies a greater decrease in future downstream per capita water availability due to population growth. The negative impact of these two socio-economic drivers outweighs the positive effects of climate change on water availability, especially under the rapid population growth of the SSP3-RCP8.5 scenario. Growing upper Indus water consumption particularly plays a substantial role in the decreasing trend of dry season water availability of the densely populated Indus plains of in the Pakistani share of the basin. Expanding water usage in the upper Indus may thus lead to *in situ* water scarcity issues in several upstream sub-basins and intensify the already considerable water stress faced in transboundary downstream areas during the dry season.

The quantified outlook on the development of upstream-downstream linkages under various drivers provided in this study holds several insights for transboundary cooperation, long-term water management and adaptation planning in the hydro-politically complex Indus basin. Foremost, adaptation strategies towards achieving the interlinked water and food security SDGs are required not just in lower Indus plains of Pakistan, but also for the Kabul and Jhelum sub-basins of the upper Indus that are administered largely by Afghanistan and India. This implies that adaptation policy and revisions of shared water management practices must explicitly consider for the impact of socio-economic changes on the evolution of upstream-downstream dependencies in the Indus basin and its transboundary implications for water demand and availability throughout it. Future disaggregated modelling assessment of the future Indus basin water system in support of these processes similarly need to include socio-economic development in the upper Indus. Subsequent research may focus on further untangling Indus upstream-downstream linkages by disaggregating hydrological dependencies within the lower Indus as well, and by evaluating implications by-and-for adaptation strategies.

**Acknowledgements**

Work of all the authors is supported by the SustaIndus project funded by NWO Wotro (Project W 07.30318.002), the Interdisciplinary Research and Education Fund (INREF) of Wageningen University and Research, and Utrecht University. HB would like to acknowledge partial funding from the Wageningen University & Research "Food Security and Valuing Water programme" that is supported by the Dutch Ministry of Agriculture, Nature and Food Security".

**Author contributions**

WS and HB conceptualised and designed the methodological approach of this study. WS collected the data, performed the data analysis and wrote the original draft paper. MJ and SD were responsible for regional validation and interpretation of model outputs. SD, AL, WI, FL, MJ and HB reviewed and edited the final draft. FL, HB and WI supervised the procedure.

**Competing interests**

The author declare no competing interests.

**Code/Data availability**

All code and data are available from the authors upon request.

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

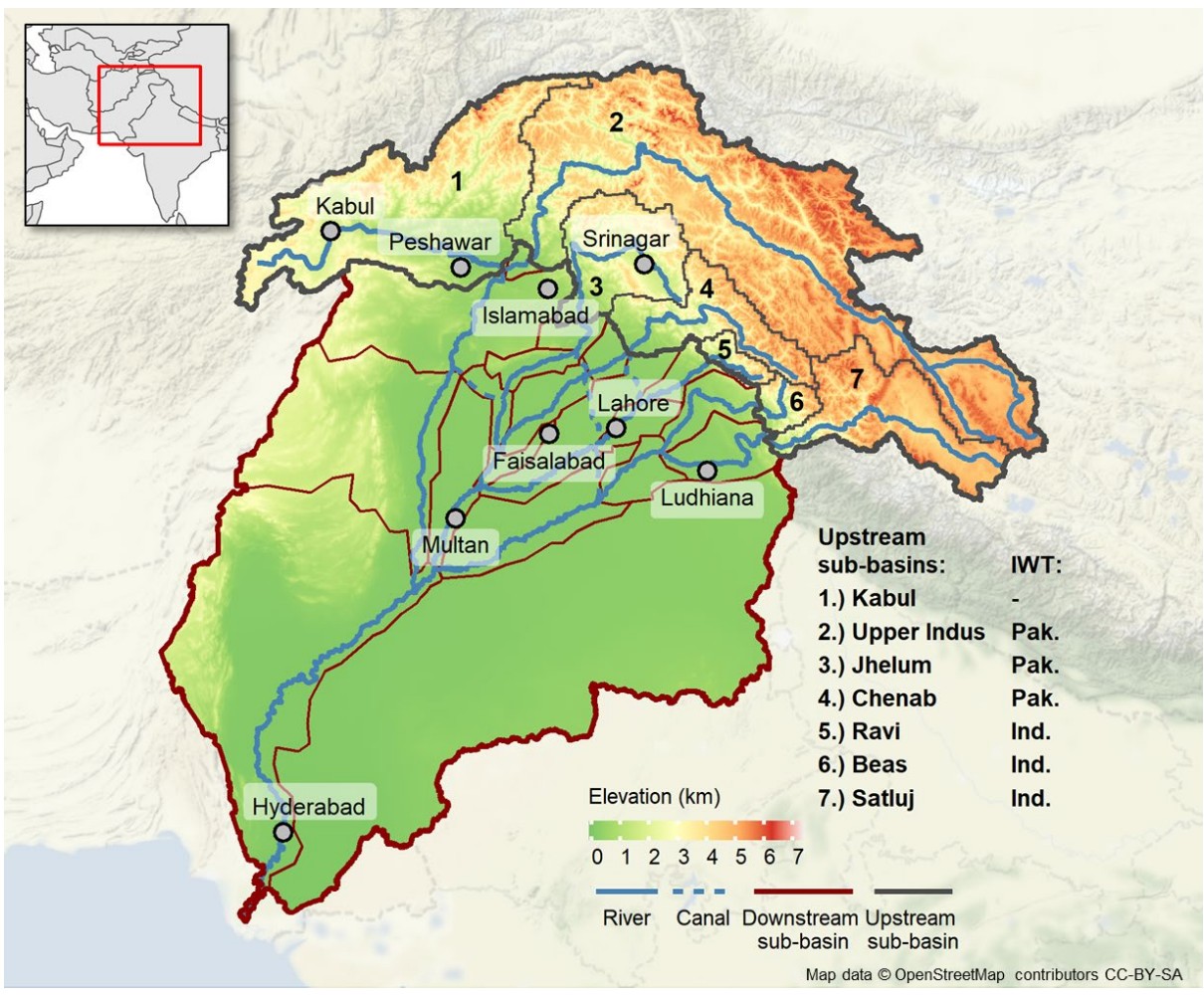

**Figure 1: Elevation map of the Indus basin with delineation of upper- (numbered) and lower Indus sub-basins, and the allotment of Indus tributaries between India and Pakistan according to the Indus Water Treaty (IWT).**


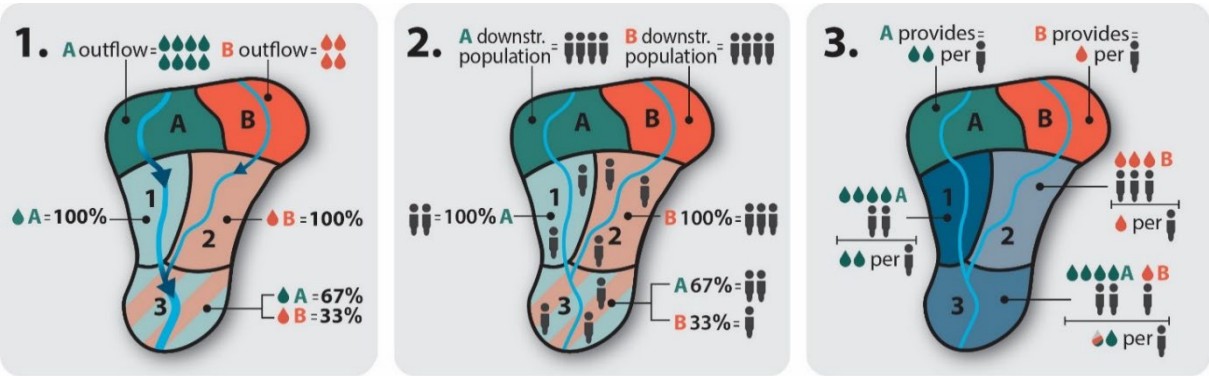

**Figure 2: Conceptual representation of the allocation of upstream sub-basin water resources to downstream sub-basins. First, (1) the relative contribution of each upstream sub-basin to the total upstream inflow of each downstream sub-basin is determined. Next, (2) the population of each downstream sub-basin is determined and assigned to the upstream sub-basins by their relative flow contribution. Lastly, (3) upstream outflows are divided by their total assigned downstream populations to obtain the per capita upstream water availability they provide to the downstream sub-basins. The upstream per capita water availability per downstream sub-basin is the mean per capita availability provided by all contributing upstream basins, weighted by their assigned populations. The total per capita water availability of a downstream sub-basin is determined by aggregating the local downstream per capita water availability and the upstream per capita water availability.**



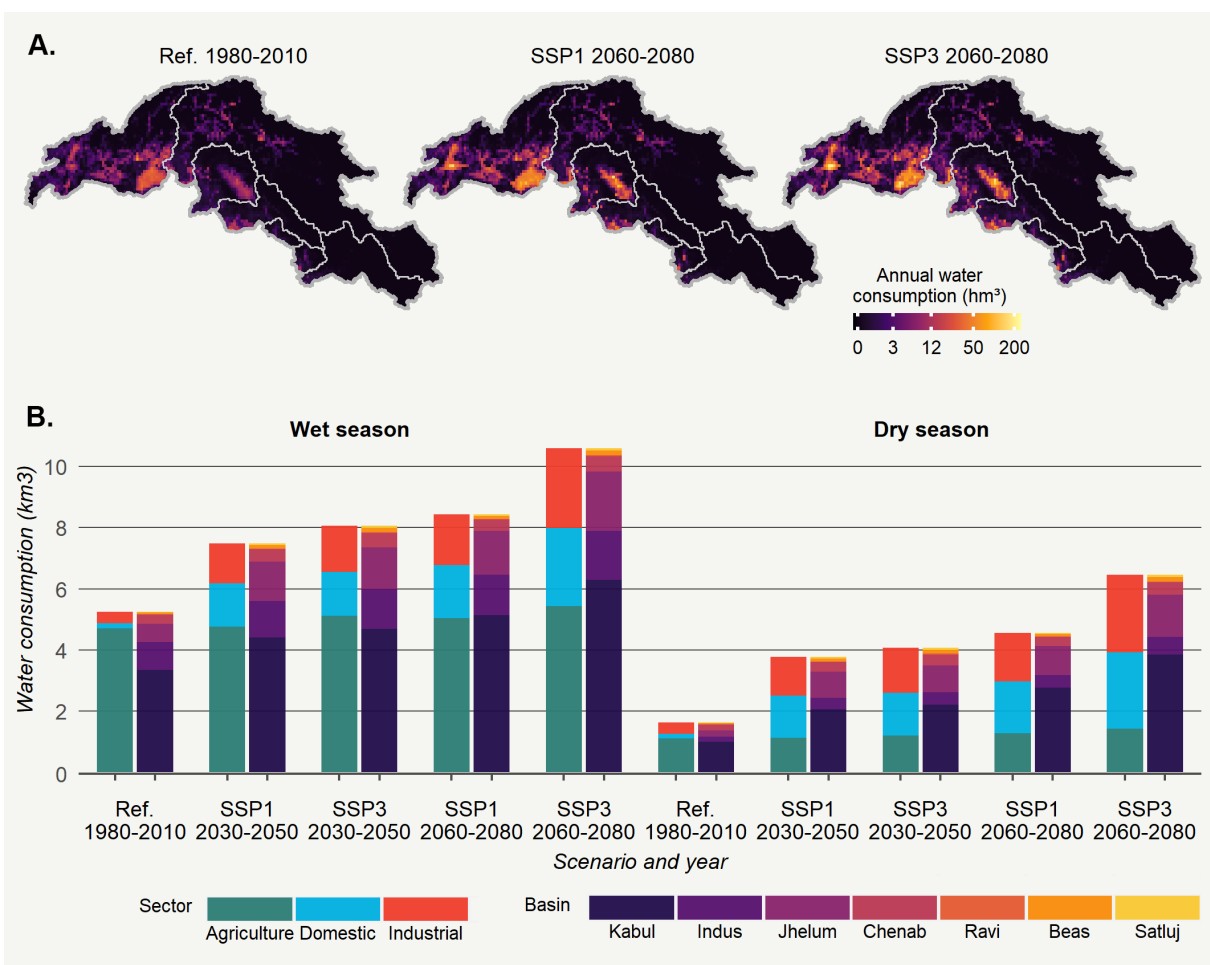

**Figure 3: Spatially (A.), seasonally and sectoral (B.) disaggregated water consumption in the sub-basins of the upper Indus basin. Agricultural water use is based on the ensemble mean. The total height of the bars (B.) indicates total water use in the upper Indus.**

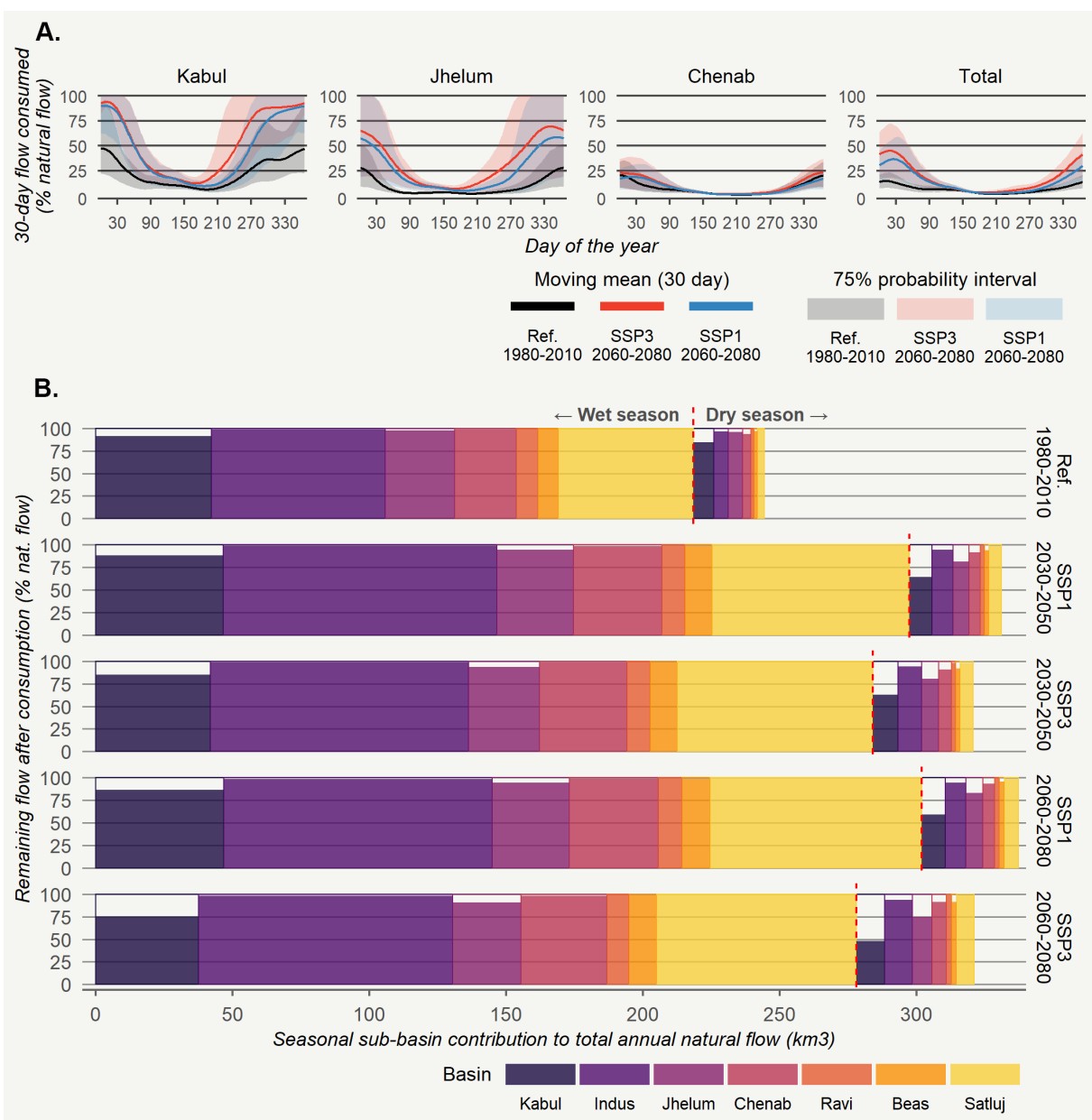

Figure 4: Daily share of natural flow consumed in upper Indus sub-basins during the reference period and the projected late time periods (A.). Development of ensemble mean absolute upper Indus outflow under climate change and the impact of consumption (B.).

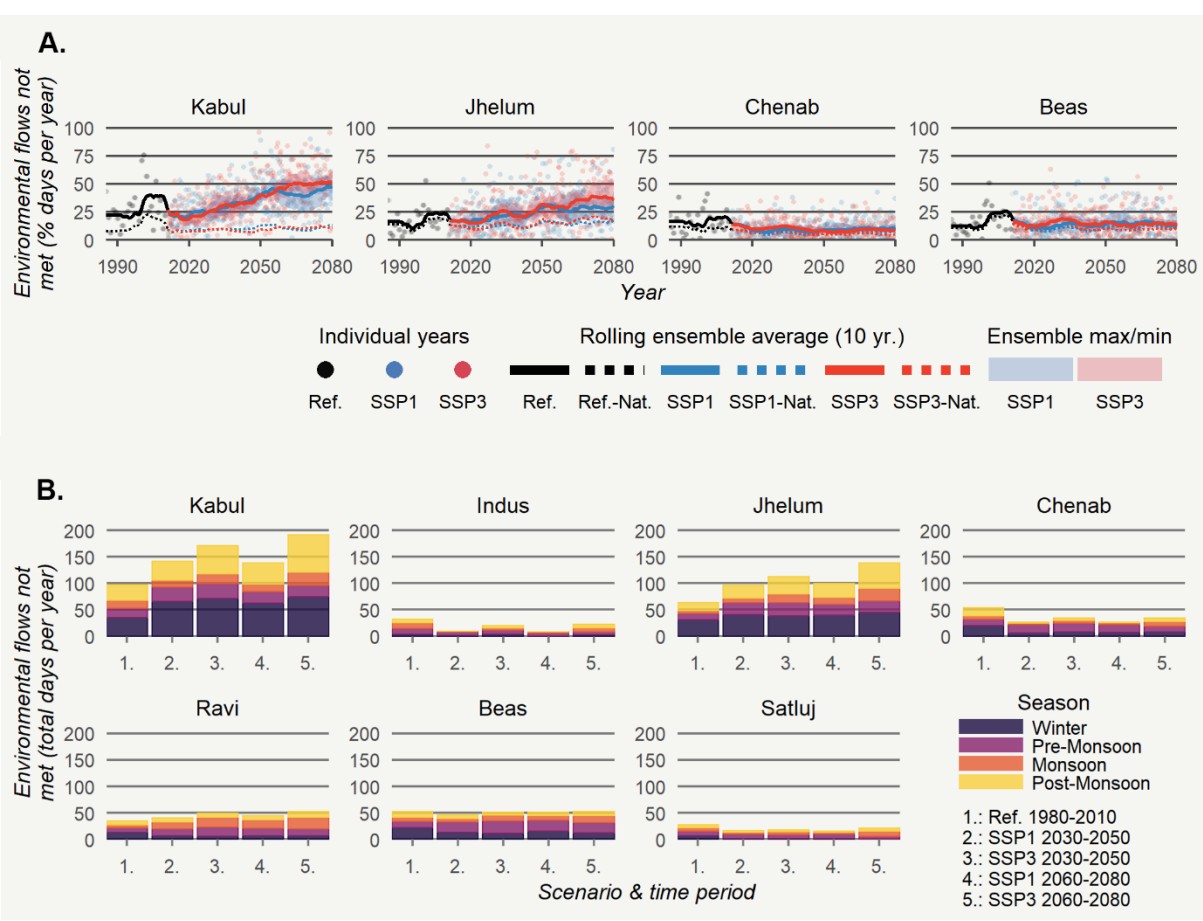

**Figure 5: Impact of upper Indus consumption on environmental flows at the outlet of the upper Indus sub basins over the assessment period (A.) and per season (B.).**

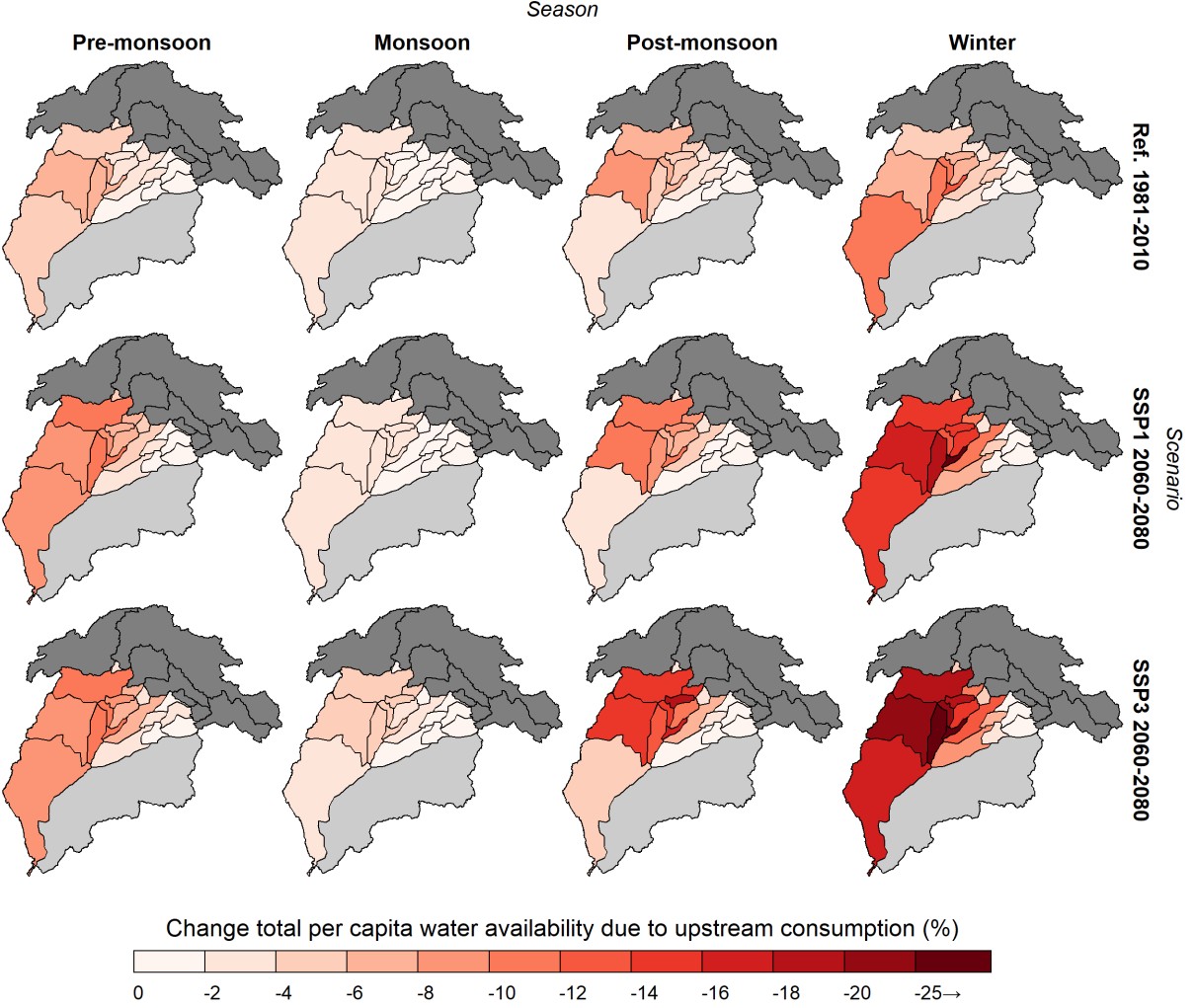

**Figure 6: Seasonal mean impact of upper Indus water consumption on the water availability per capita of the lower Indus sub basins for all years and ensemble members. The dark grey area herein represents the upper Indus sub-basins. The light grey area is not downstream of any of the upper Indus sub-basins and is therefore omitted.**


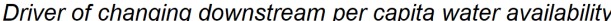

*Driver of changing downstream per capita water availability*

**Climate change**   **Downstream population change**   **Upstream consumption**   **All drivers**

*Season and scenario*

Wet Season — SSP1

Dry Season — SSP1

Wet Season — SSP3

Dry Season — SSP3

Total change (%) downstream per capita water availability

← -60   -50   -40   -30   -20   -10   -5   0   20   50 →

**Figure 7: Isolated impact of climate change, downstream population change and upstream consumption on seasonal lower Indus water availability in the late period (i.e. compared to late period situation without the effect of the respective driver). Additionally the change in late period water availability with all drivers considered, compared to reference period water availability.**


**Table 1: Input datasets used for water accounting analysis**

| Input dataset | Resolution (time/space) | Source |
|---|---|---|
| *Discharge* | | |
| Upper Indus | Daily 1980-2100/ Sub-basin outlets | Wijngaard et al. (2017) |
| Lower Indus | Daily 1980-2080/ 5 arcmin | Simulated by this study, model and calibration from Bondeau et al. (2007) & Biemans et al. (2016) |
| *Consumption* | | |
| Domestic | Annual 1980-2080/ National level | Simulated by this study, model and calibration from Bijl et al. (2016) |
| Industrial | Annual 1980-2080/ National level | Simulated by this study, model and calibration from Bijl et al. (2016) |
| Agricultural | Monthly 1980-2080/ 5 arcmin | Simulated by this study, model and calibration from Bondeau et al. (2007) & Biemans et al. (2016) |
| *Scenarios* | | |
| Population projections | Annual 1980-2080/ 5 arcmin | Smolenaars et al., (2021) for future (2015-2080) & Klein Goldewijk et al. (2011) for historical (1980-2015) |
| GDP projections | Annual 1980-2080/ National level | Future (2015-2080) Smolenaars et al., 2021 & historical (1980-2015) Dellink et al. (2017) |
| Climate data | Daily 1980-2100/ 5 arcmin | Lutz, ter Maat, et al. (2016) |

**Table 2: Development of population, water consumption, natural flow and water availability (ensemble means) for the upper Indus sub-basins (relative change between brackets) for the reference (1980-2010) mid (2030-2050) and late (2060-2080) period. In the water availability columns, the occurrence of water stress (m3/cap/year < 5000m3) in a sub-basin is indicated by providing the values in *italics*. Moderate water stress (m3/cap/year < 1500m3) is additionally indicated with \* and severe water stress (m3/cap/year < 1000m3) with \*\*.**

| Sub-basin | Population (millions) | | | | | Natural flow (km3) | | | | | | | | | |
|---|---|---|---|---|---|---|---|---|---|---|---|---|---|---|---|
| | *Ref.* | *Mid.* | | *Late.* | | *Ref.* | | *Mid.* | | | | *Late.* | | | |
| | - | *SSP1* | *SSP3* | *SSP1* | *SSP3* | - | | *SSP1* | | *SSP3* | | *SSP1* | | *SSP3* | |
| | | | | | | *Dry* | *Wet* | *Dry* | *Wet* | *Dry* | *Wet* | *Dry* | *Wet* | *Dry* | *Wet* |
| Kabul | 16 | 40 (150%) | 47 (194%) | 46 (188%) | 74 (363%) | 7.5 | 42 | 8.3 (11%) | 47 (12%) | 9.2 (23%) | 42 (0%) | 8.7 (16%) | 47 (12%) | 10 (33%) | 37.7 (-10%) |
| Upper Indus | 4.5 | 6.9 (53%) | 8.1 (80%) | 6.2 (38%) | 9.4 (109%) | 5.2 | 63 | 7.8 (50%) | 100 (59%) | 8.7 (67%) | 94 (49%) | 7.4 (42%) | 98 (56%) | 10 (92%) | 93 (48%) |
| Jhelum | 7.9 | 16.9 (113%) | 17.1 (116%) | 16.6 (110%) | 23.1 (192%) | 5.3 | 25 | 5.9 (11%) | 28 (12%) | 6.2 (17%) | 26 (4%) | 6.4 (21%) | 28 (12%) | 7.2 (36%) | 25 (0%) |
| Chenab | 2.6 | 3.9 (50%) | 4.4 (69%) | 3.0 (15%) | 4.7 (81%) | 3.0 | 23 | 4.1 (37%) | 32 (39%) | 4.6 (53%) | 32 (39%) | 4.3 (43%) | 33 (43%) | 5.2 (73%) | 31 (35%) |
| Ravi | 0.33 | 0.26 (-21%) | 0.41 (24%) | 0.11 (-66%) | 0.31 (-6%) | 1.1 | 7.9 | 1.5 (36%) | 8.4 (6%) | 1.6 (45%) | 8.4 (6%) | 1.6 (45%) | 8.6 (9%) | 1.8 (64%) | 8.2 (4%) |
| Beas | 0.95 | 1.4 (47%) | 1.7 (79%) | 1.4 (47%) | 2.0 (110%) | 1.3 | 7.6 | 1.7 (31%) | 10 (32%) | 1.7 (31%) | 9.9 (30%) | 1.9 (46%) | 10 (32%) | 1.9 (46%) | 10 (32%) |
| Satluj | 0.68 | 0.82 (20%) | 1.1 (62%) | 0.58 (-15%) | 1.2 (76%) | 2.5 | 49 | 4.4 (76%) | 72 (47%) | 4.7 (88%) | 71 (45%) | 5.2 (108%) | 77 (57%) | 6.4 (156%) | 73 (49%) |
| **Total** | **33** | **70 (112%)** | **80 (142)** | **74 (124%)** | **114 (245%)** | **26** | **218** | **34 (31%)** | **297 (36%)** | **37 (42%)** | **284 (30%)** | **35 (35%)** | **302 (39%)** | **43 (65%)** | **278 (28%)** |

| Sub-Basin | Water consumption (km3) | | | | | Water availability (m3/cap/year) | | | | |
|---|---|---|---|---|---|---|---|---|---|---|
| | *Ref.* | *Mid.* | | *Late.* | | *Ref.* | *Mid.* | | *Late.* | |
| | - | *SSP1* | *SSP3* | *SSP1* | *SSP3* | - | *SSP1* | *SSP3* | *SSP1* | *SSP3* |
| Kabul | 4.3 | 6.5 (51%) | 6.9 (60%) | 7.9 (84%) | 10 (135%) | *3090* | *1380\* (-55%)* | *1090\* (-65%)* | *1210\* (-61%)* | *640\*\* (-79%)* |
| Upper Indus | 1.1 | 1.6 (45%) | 1.7 (55%) | 1.7 (55%) | 2.2 (100%) | 15160 | 15620 (3%) | 12680 (-16%) | 17000 (12%) | 10960 (-28%) |
| Jhelum | 0.81 | 2.1 (159%) | 2.3 (184%) | 2.4 (196%) | 3.3 (307%) | *3840* | *2010 (-48%)* | *1880 (-51%)* | *2070 (-46%)* | *1390\* (-64%)* |
| Chenab | 0.48 | 0.74 (54%) | 0.83 (73%) | 0.67 (40%) | 0.91 (90%) | 10000 | 9260 (-7%) | 8320 (-17%) | 12430 (24%) | 7700 (-23%) |
| Ravi | 0.03 | 0.05 (35%) | 0.066 (91%) | 0.03 (-3%) | 0.06 (71%) | 27270 | 38080 (40%) | 24390 (-11%) | 92730 (240%) | 32260 (18%) |
| Beas | 0.09 | 0.19 (111%) | 0.23 (156%) | 0.18 (100%) | 0.29 (222%) | 9370 | 8360 (-11%) | 6820 (-27%) | 8500 (-9%) | 5950 (-36%) |
| Satluj | 0.05 | 0.11 (104%) | 0.15 (178%) | 0.09 (70%) | 0.17 (215%) | 75740 | 93170 (23%) | 68820 (-9%) | 141720 (87%) | 66170 (-13%) |
| **Total** | **6.9** | **11 (64%)** | **12 (77%)** | **13 (88%)** | **17 (146%)** | **7380** | **4720 (-36%)** | **4010 (-46%)** | **4560 (-38%)** | **2790 (-62%)** |

| Sub-Basin | Water availability- only pop. change (m3/cap/year) | | | | | Water availability – only climate change (m3/cap/year) | | | | |
|---|---|---|---|---|---|---|---|---|---|---|
| | *Ref.* | *Mid.* | | *Late.* | | *Ref.* | *Mid.* | | *Late.* | |
| | - | *SSP1* | *SSP3* | *SSP1* | *SSP3* | - | *SSP1* | *SSP3* | *SSP1* | *SSP3* |
| Kabul | *3090* | *1240\* (-60%)* | *1050\* (-66%)* | *1080\* (-65%)* | *670\*\* (-78%)* | *3090* | *3460 (12%)* | *3200 (4%)* | *3480 (13%)* | *2980 (-4%)* |
| Upper Indus | 15160 | 9880 (-35%) | 8420 (-44%) | 11000 (-27%) | 7260 (-52%) | 15160 | 23960 (58%) | 22820 (51%) | 23420 (54%) | 22890 (51%) |
| Jhelum | *3840* | *1790 (-53%)* | *1770 (-54%)* | *1830 (-52%)* | *1310\* (-66%)* | *3840* | *4290 (12%)* | *4080 (6%)* | *4350 (13%)* | *4080 (6%)* |
| Chenab | 10000 | 6670 (-33%) | 5910 (-41%) | 8670 (-13%) | 5530 (-45%) | 10000 | 13880 (39%) | 14080 (41%) | 14350 (44%) | 13920 (39%) |
| Ravi | 27270 | 34620 (27%) | 21950 (-20%) | 81820 (200%) | 29030 (6%) | 27270 | 30000 (10%) | 30300 (11%) | 30910 (13%) | 30300 (11%) |
| Beas | 9370 | 6360 (-32%) | 5240 (-44%) | 6360 (-32%) | 4450 (-53%) | 9370 | 12320 (31%) | 12210 (30%) | 12530 (34%) | 12530 (34%) |
| Satluj | 75740 | 62800 (-17%) | 46820 (-38%) | 88790 (17%) | 42920 (-43%) | 75740 | 112350 (48%) | 111320 (47%) | 120880 (60%) | 116760 (54%) |
| **Total** | **7380** | **3470 (-53%)** | **3050 (-59%)** | **3290 (-55%)** | **2120 (-71%)** | **7380** | **10050 (-53%)** | **9710 (-59%)** | **10230 (-55%)** | **9720 (-71%)** |

 **Appendix**

$$C_r^I = V_r(t) * \alpha * G_r(t)^b * R_r^I * F_r^I * E_r^i(t) \qquad (1)$$

$$C_r^M = P_r(t) * \frac{c}{1+e^{\left(\frac{m-\ln(G_r(t))}{s}\right)}} * R_r^M * F_r^M * E_r^M(t) \qquad (2)$$

**Appendix 1: Formulas to determine industrial (1) and municipal (2) water consumption from Bijl et al. (2016) whereby *C* stands for the consumption (m³/yr) for the industrial (*I*) and municipal (*M*) sector for year *t* and region *r*. The models first determine the structural withdrawals for a region in a year, for which *V* is the economic driving force of total industry value added (US$/yr), *P* is the population (yr), *G* is the is the level of economic development (expressed in $US GDP per capita/yr). These are then multiplied by a static region factor (*R*) that accounts for cultural factors, a static consumption fraction (F) and an annual efficiency factor (*E*). The industrial model moreover has two parameters, *α* and *b*, that were calibrated at 3.57 and −0.564 respectively. The municipal model contains two parameters, *c*, and *s*, that were calibrated at 8.575 and 0.6985 respectively. Additionally a midpoint (*m*) was defined at 143.5 (m³/cap/yr) by Bijl et al. (2016). The economic and population data to run these models were sourced from Smolenaars et al. (2021) and are described in the methodology. The region factors, consumption fraction and efficiency factors were sourced from Bijl et al. (2016).**

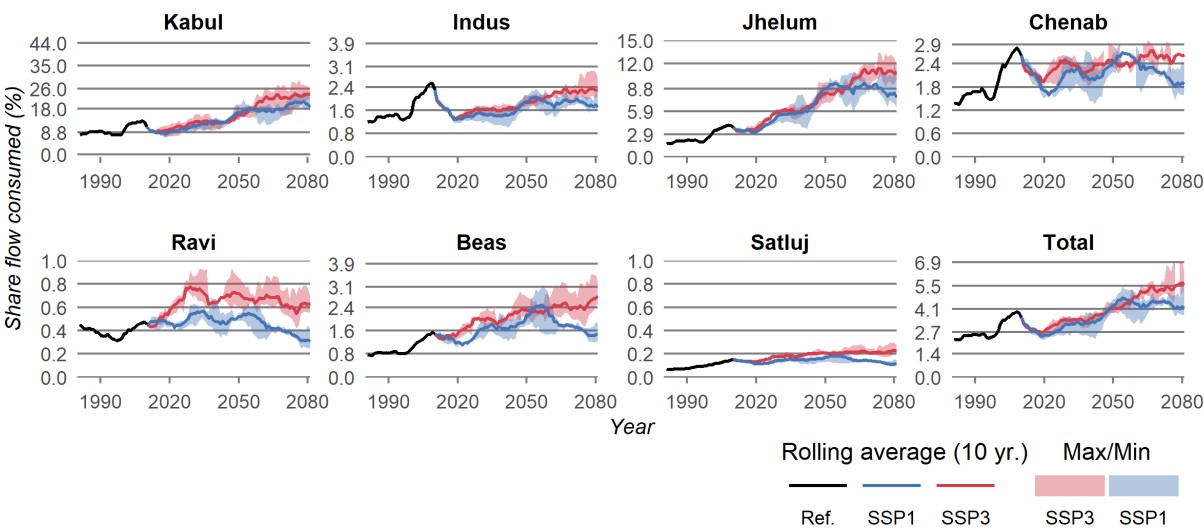

**Appendix 2: share of annual discharge consumed per sub-basin and for the total upper Indus basin**

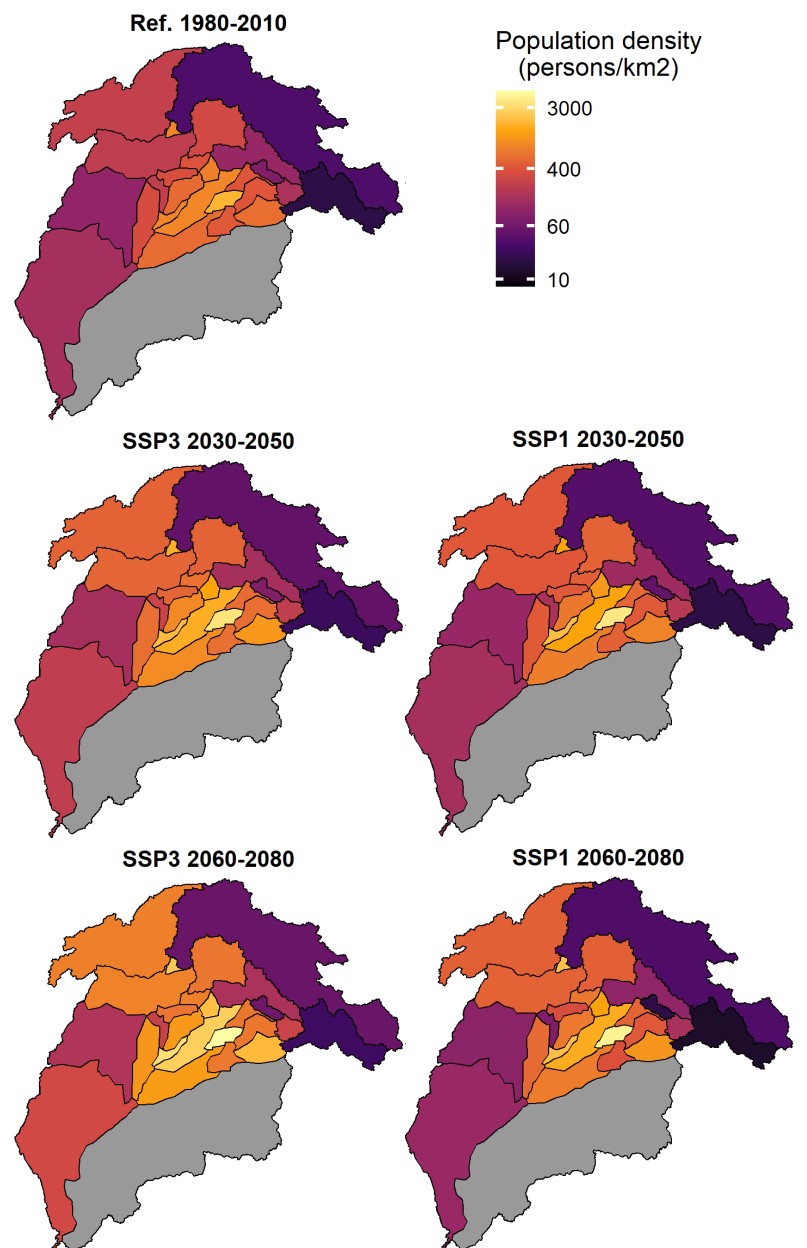


**Appendix 3: Population density of each upper and lower Indus sub-basin through time and for both scenarios, as used in this study.** The population projections were sourced from Smolenaars et al (2021, under review). They were developed by spatially downscaling the national population projections of the global SSP framework using regionalized population model that considers for urbanization, internal highland-to-lowland migration and proximity to infrastructure. These drivers were weighted relative to the 770 scenario context sourced from both the global SSPs and pre-existing qualitative regional development storylines developed by Roy et al. (2019).

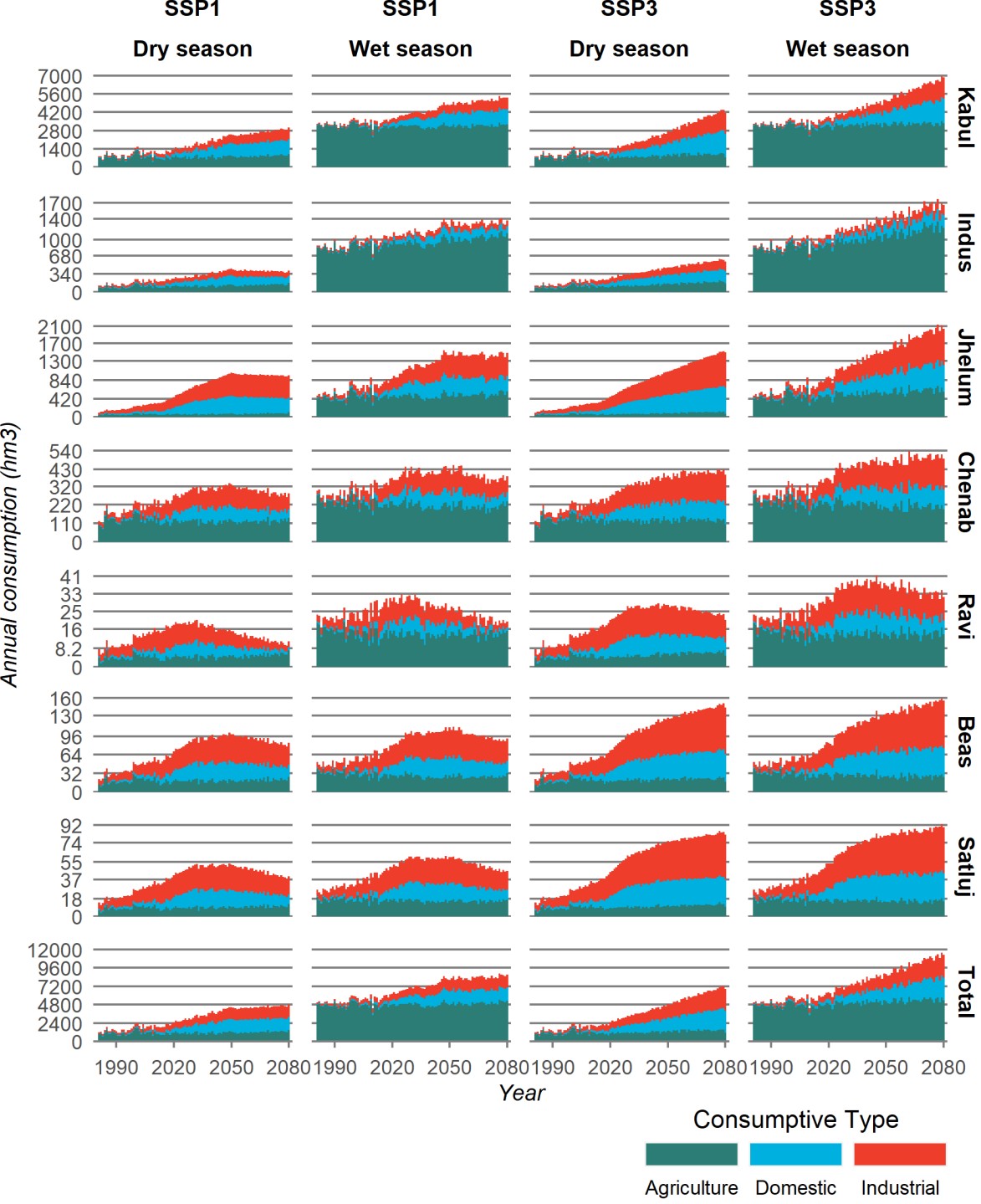

**Appendix 4: Domestic, industrial and agricultural water development per season and scenario.**

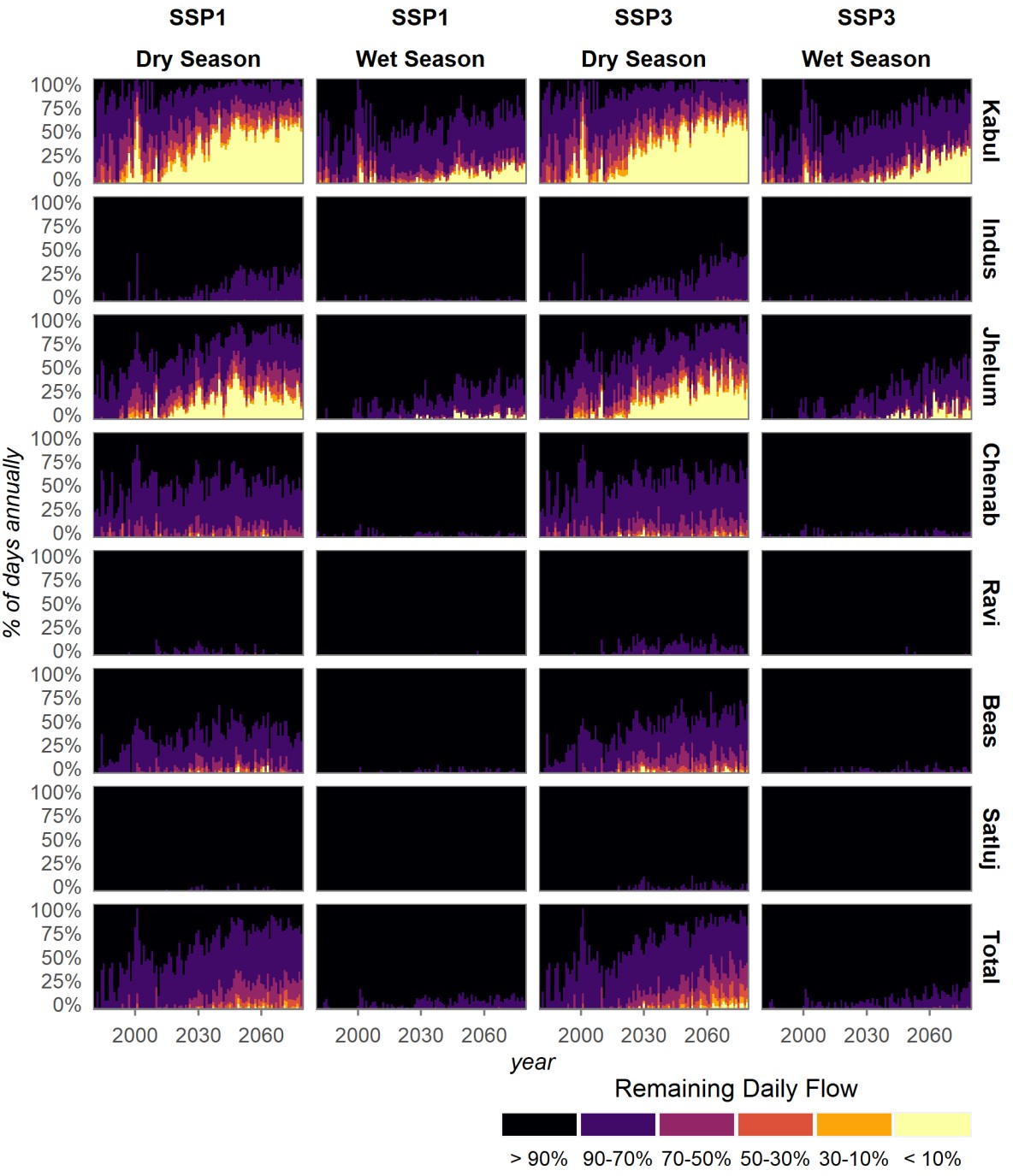

**Appendix 5: Development of the daily remaining flow per season and per scenario.**