# Peer review of "Future upstream water consumption and its impact on downstream water availability in the transboundary Indus basin"

_Hydrology and Earth System Sciences, 2021_

## Author Comment (AC1)

**Response to Reviewer 1**

First of all we would like to thank the reviewer for the constructive comments and suggestions. Hereby, we would like to provide a point-by-point reply for each of the comments in the table below. We are confident that the reviewers comments and corresponding suggested changes will increase the quality of the article. The reviewer's original comments are numbered and in italics for clarity. References which do not appear yet in the main manuscript are listed at the end of this document. We aim to modify the manuscript according to our responses.

1.) *This study cannot be defined as socio-hydrology research; however, their focus is still on a transboundary river.*

**Author response:** We acknowledge that in our research the socio-hydrological concept is not explicitly mentioned. However, we would like to argue that our study does fall within the scope of socio-hydrology. Following the definition by Sivapalan et al. (2011), socio-hydrology can be defined as the study of the dynamic interactions in coupled human-water systems. Our study fits well within this domain as it aims to better understand the impacts of socio-economic changes in the upper Indus on its future hydrology, and consequently, the impact this interaction has on downstream water availability for human uses. We assess this human-water systems interaction by considering both hydro-climatic and socio-economic drivers (population change, economic growth, urbanisation). Our study provides an innovative understanding of; 1.) the upstream-downstream linkages within the transboundary Indus water system and 2.) the role of these drivers herein, both of which are to some extent socio-hydrological in character. Moreover, according to Sivapalan et al., (2011) "*socio-hydrology must strive to be a quantitative science. While broad narratives may be important for context, quantitative descriptions are needed for testing hypotheses, for modelling the system and for predicting possible future trajectories of system states.*". In light of the last two points, we use a quantitative modelling approach to simulate potential system states, and we use this to improve our understanding of the future trajectories of upstream-downstream linkages (line 342-370) as indicated in line 74-75 of the introduction. To make the relation to socio-hydrology of our study clearer, we will expand this part of the introduction and highlight that our approach does contain socio-hydrological relevance.

2.) *The authors claimed that their novelty is to reveal the effect of upstream water consumption in the Indus water availability under climate change and population growth. However, it is not clear if they have any other innovation in their approach compared to general research in transboundary rivers.*

**Author response:** The innovation and novelty in our article, compared to previous quantitative transboundary research, stems from the following factors;

- In terms of research outcomes, we provide in our article a first time regionalised quantification of the effects of climatic and socio-economic drivers on future upstream-downstream (and thus transboundary) linkages in the Indus basin. Previous quantitative studies of transboundary upstream-downstream dependencies that include the Indus have largely been done at the global scale. These studies made coarse, aggregated assessments, using global parameters, dividing the basin into an upstream and downstream sub-basin (Viviroli et al., 2020) or an upstream, midstream and downstream sub-basin (Munia et al., 2016; Munia et al., 2018; Munia et al., 2020). However, the upper Indus basin is highly diverse in terms of hydrology and the degree of anthropogenic influence on the water system. Moreover, it is administrated by four different states. Contrary to coarse global studies, our regional

[Figure]

*Figure 1: downstream area of each upper Indus sub-basin, including those areas connected through linkage channels.*

study assesses the trends and impacts of changing upstream consumption at a higher resolution, involving seven upper Indus sub-basins, and eighteen lower Indus sub-basins that all receive upstream water from a different combination of upper Indus sub-basins. This increase in resolution of our study thus allows for a much more detailed quantitative understanding of future upstream-downstream dependency hotspots within the Indus basin, whereas previous global assessments only provided insight into the upstream-downstream dependency of the entire Indus basin at large. From a transboundary perspective, our study therefore also provides new quantified insights into how upstream-downstream linkages in sub-basins shared between multiple riparian states (e.g. Kabul basin; Afghanistan-Pakistan, and Jhelum and Chenab basins; India-Pakistan) are potentially affected by the drivers of interest. This allows us to draw novel conclusions on where and when future transboundary hydropolitical tensions are likely to occur in the Indus basin, in a way that the lumped representation of existing transboundary water assessments was not able to.

- To obtain these outcomes, we developed a simple yet novel approach for quantifying upstream-downstream linkages in complex transboundary basins with variable water flow directions. The aforementioned (global) studies that quantify upstream-downstream dependencies relied on frameworks that assume that each basin consists of two or three coarse sub-basins with a linear connection based on natural flow direction. However, in reality, the spatial allocation of upper Indus water over downstream areas is highly controllable, due to an expansive network of barrages and linkage channels (Wescoat Jr et al., 2018). This infrastructure allows riparian states to optimally distribute the water of the Indus tributaries allotted to them by the Indus Water Treaty (see Basharat, 2019). The availability of upstream water to downstream sub-basins is hence variable, and this non-linearity cannot be simulated with fixed natural flow approaches used in previous studies. Our new approach therefore determines water availability directly on a per-capita basis by allocating surplus water resources of upstream sub-basins equitably over all their downstream inhabitants, including those in areas connected through linkage channels (see figure 2 in the manuscript). This allows us to jointly assess multiple upper Indus sub-basins with overlapping and converging downstream areas (see Figure 1 of the response), even if those areas are shared by multiple riparian states, and without having to make quantitative assumptions as to how water is shared between competing sub-basins.

Additionally, the effects of downstream population changes, both in total amounts and in spatial distribution, on transboundary upstream-downstream dependencies can be assessed using a transparent approach, which is explored in Figure 7 of the manuscript. As such, the approach developed in this study can also be used in other complex transboundary basins for high-resolution quantifications of upstream-downstream linkages as well as identify potential hotspots for water conflicts.

Upon revisiting our manuscript, we acknowledge that the explanation of our novel approach and findings, from a transboundary perspective, should be emphasised more. We therefore suggest several additions to our manuscript:

- First, we will add a section in the introduction discussing the findings and limitations of existing global transboundary upstream-downstream studies (and their approaches) for the Indus. This section will highlight why the knowledge provided by existing (global) studies is insufficient in relation to transboundary water management in the Indus basin. We intend to add this segment throughout the 3th, 4th and 5th paragraph of the introduction (line 46-68) and base it on the explanation in the first bullet-point above.
- Secondly, we will improve the explanation for the approach used in our study in the 'Methods and Materials' section and highlight why our methodological choices aid in our quantitative assessment of future transboundary upstream-downstream linkages. We will do this in section 2.2 'Sub-basin delineation' by explaining our division of the upper Indus into multiple sub-basins, which is different from previous lumped approaches. We will also expand the explanation in 'Analysis and data sources' section (line 191-192) on the logic behind our population based water allocation approach for the lower Indus. Here, we will specifically explain the transboundary relevance of allowing variable water allocation and of including linkage channels. This section will be based on the clarifications in the second bullet point above. In addition, we will stress in the introduction that the population based water allocation approach we developed is a study outcome that is applicable to general transboundary research at the regional level (line 75).
- Lastly, within the discussion section dedicated to our approach (line 330-355) we will add a reflection upon the broader implications and applicability of our approach for future transboundary upstream-downstream studies in complex basins. In addition, we add our reflection on the difference between existing lumped and our novel disaggregated assessment of the upper Indus (line 345-350) and its transferability to general transboundary upstream-downstream research.

*3.) Although the authors chose a transboundary river for their analysis, I am surprised how this research explicitly adds to each countries' understanding of their shared water resources in the future.*

**Author response:** Our research results provide insight into multiple changes in upstream water consumption and downstream availability that are important for future transboundary water management in the Indus basin (kindly refer also to the response to comment 2 that addresses this further). In the current version of the manuscript, we specifically highlight (line 361-370) how adaptive measures in the Kabul and Jhelum sub-basins (administered largely by Afghanistan and India, respectively) are likely required to deal with increasing water stress, but may also affect the availability of water in the in the expanding urban centres of downstream Pakistani. In this way, our estimates provide insights into the impact of changing water consumption patterns, constrained by population growth and climate change, on each individual country and between the riparian states. From a basin perspective, this allows us to identify hotspots where transboundary upstream-downstream impacts may increase, and consequently where hydropolitical tensions between riparian states may occur or escalate. Moreover, although the role of climate change in

Indus basin water management and water sharing treaties has been at the forefront of scientific attention (Qamar et al., 2019; Ali & Bhargava, 2021), the decoupling of climate and socio-economic drivers in our study demonstrated that socio-economic changes may have a more prominent influence on future upstream-downstream linkages. Our assessment thus illustrates that any course of action towards shared future water management must explicitly account for socio-economic changes, both in magnitude as in spatial distribution, alongside the effects of climate change.

To place more emphasis on these transboundary consequences, and on their implications for international water management in the basin, we suggest to make four changes to our manuscript:

- First, the present results section spatially presents research outcomes at the sub-basin level. We have chosen this bio-physical spatial unit in our figures to keep a scientifically neutral stance, given the present hostilities between riparian states. However, we do acknowledge that this limits the insight our study provides for individual countries. We will therefore describe in what riparian states changes and impacts are occurring in the accompanying text of figures in the results section, without explicitly adding this information in a quantitative manner to the maps and figures.
- Secondly, the present reflection on upstream-downstream impacts in the discussion section (line 342-352) is largely transboundary in nature, since both the Jhelum and Kabul sub-basins are administered by multiple riparian states. However, this section currently does not make these transboundary implications clear, as the upstream-downstream relations are described using geographical and sub-basin names. We will therefore add specific references the riparian states involved in these changing upstream-downstream linkages, similar to naming convention used in line 361-370. We will add clarifications on the implications for individual riparian states as well in the section that describes the patterns of increasing water stress in upper Indus sub-basins (line 353-360).
- Thirdly, we will expand the discussion on the implications of our study for shared water management between the riparian states and the context of existing treaties (line 362-370). This section will be aided by an expanded explanation of present day water management and water sharing treaties that we will add after the introduction (we refer here to the response to comment 7).
- Lastly, the present key messages and lessons-learnt of our paper in the discussion (line 371-375) and the conclusion (line 393-402) focus largely on the implications of changing upstream-downstream dependencies for the pursuit of water and food security SDGs, and on future hydrological modelling studies of the Indus basin. We will expand these segments to provide key highlights for transboundary water management as well, based on the clarifications above.

*4.) Lines 1-2: downstream water availability.*

**Author response:** We agree that the clarity of the title improves with this correction and will change it accordingly in the next version of the manuscript.

*5.) The introduction lacks literature on transboundary rivers in general. The authors should then explain if they have any novelties compared to the previous studies in terms of methodology. The focus of the current introduction is too much on the case study. Also, there is nothing about the approach used by this study and its comparison with approaches in previous studies.*

**Author response:** The main topic of our research is the Indus basin and the implications that changing upstream water use may have here for transboundary downstream water availability. In this sense, our research is a regional transboundary assessment more so than a case study for a broader approach. We

acknowledge that the broader context of transboundary science holds importance for the Indus basin as well, but we do prefer to keep the introduction with the basin itself and not with transboundary rivers in general. We therefore propose to make several changes to our introduction to place our study in the larger context of existing quantitative transboundary water management research outcomes and approaches. We refer here to the response to comment 2 in which we discuss in depth the proposed updates to the introduction and cited literature therein. Moreover, we will also describe the novelty of our approach, as compared to the approach taken by previous studies, in the introduction and methodology sections, and reflect on the transferability of this approach for general upstream-downstream research in the discussion. We kindly refer to the more detailed explanation of these changes in the responses to comments 2, 7 and 8.

6.) *Lines 53-55: How do the authors can compare their work with studies by Amin et al. (2018) and Mehbood and Kim (2021) in terms of their approaches? What is the authors' argument for not taking a similar approach with these studies?*

**Author response:** Previous regional studies have not provided any insight into downstream (and thus transboundary) water management implications of upstream changes within the basin, which is the main objective of our article. In comparison to existing regional studies into future upper Indus water use, our study goes beyond the scope of research by Amin et al. (2018) and Mehbood & Kim (2021), as is explained in lines 54-55. These studies only assessed future water demand and consumption in the Pakistani share of the upper Indus. The downstream implications of changing upper Indus consumption patterns are hence not considered. Moreover, it also exceeds the work of Biemans et al. (2019) that looked at the dependency of the lower Indus on mountain water, but did so without considering for upstream consumption. We will further clarify this difference in scope between our study and previous regional studies within the introduction.

7.) *After the introduction section, this study sharply goes to the method. A section needs to introduce the case study with a map of the basin, its hydrologic condition, water supply/demand sites, and other details. As this paper is about a transboundary river, I would suggest the authors also write about transboundary agreements currently/historically used in the basin and show countries' water rights.*

**Author response:** We thank the reviewer specifically for this suggestion and we acknowledge that the manuscript would benefit from an expanded overview of present transboundary water management practices. We specifically suggest to add a 'Current Water Management and Treaties' section at the start of the 'Methods & Materials' chapter. In this section we will specifically discuss the present day hydrological and water management status of the Indus basin from a transboundary perspective. Here, we will also explain the role of the Indus Water Treaty and the Pakistani Water Apportionment Accord. Figure 1 of the manuscript will also be introduced in this section and will be modified to additionally show relevant information on existing treaties/water rights.

8.) *I would suggest authors use separate sections for data and scenarios.*

**Author response:** The present 'Materials & Methods' section contains a separate section for the scenarios, while most data is introduced within the 'Analysis and data sources' segment. However, the scenarios that we use in our study are quantitative and spatially explicit and this has caused some overlap to occur between the 'narrative' and 'forcing data' aspects of the scenarios. Upon revisiting the document we do

understand that the current blended introduction of scenario narratives and forcing data makes it more difficult to understand the link between the scenarios and our analysis.

We suggest the following changes to remediate this:

- Primarily, we will restructure the 'Methods & Materials' section. We will do so by reducing the 'Spatially explicit scenario context' section (line 84-100) to only contain the qualitative, narrative-related information of the scenarios. The forcing data information of the scenarios will be moved to the beginning of the 'Analysis and data sources' section. Here. it will form sub-section '2.3.1. Scenario forcing data', which explains the input data that was used to generate the actual hydrological and water consumption data that was used directly in our water accounting assessment.
- Additionally, we will add proper references to Table 1 within the 'Analysis and data sources' section to aid in the understanding of what data was exactly used in each part of the analysis. We will adapt the terminology in this table to be consistent with that used in the text.
- Lastly, we will revise the general overview of our general approach of our methodology (lines 78-83) to more clearly outline the division between scenario, data and analysis.

*9.) Lines 78-83: The purpose of these lines is not clear. The authors just explain some incomplete information about the work and just focus on the scenarios.*

**Author response:** The purpose of these lines is to provide a summary introduction of the general methodology and an overview of the structure of the 'Methods & Materials' section. We will revise this paragraph in accordance to the responses to comments 2 and 8. In addition, we will more clearly introduce the structure of the methodology in this segment.

*10.) Table 1 is not discussed in the manuscript.*

**Author response:** We will make references to Table 1 in the 'Methods & Materials' overview (line 78-83) and in the 'Analysis and Data Sources' section.

*11.) Line 121: what does "SPHY" stand for?*

**Author response:** SPHY stands for "Spatial Processes in Hydrology". We will clarify this in line 121.

*12.) I cannot find the Annex in general in the submission.*

**Author response:** Our apologies for the glitch in accessing our Annex figures. To avoid this in future submissions, we will be directly appended the annexes to the end of the manuscript.

*13.) Line 161: "blue water consumption" should be defined as it is mentioned for the first time in the manuscript.*

**Author response:** We will add our definition of blue water consumption ("water resources that are extracted from surface water or groundwater and are subsequently evaporated or lost i.e. taken out of the water balance of the subbasin") in line 161.

*14.) The result section should be organized in the way that each table/figure can be discussed in one subsection. It would read better than the current format.*

**Author response:** The present result section is organised along three subsections that each present a separate aspect of our study. We believe that the figures and text that together form a subsection are complementary and that important information may be lost if these are discussed completely separated (e.g. see response to comment 16). We will however make changes to the order in which information is presented within these subsections to improve readability and flow of information.

*15.) A general point in the result section: the authors always use numbers that are not clearly shown in the tables and figures. If the authors prefer to describe the numbers in terms of percentage, it is better to have a column in table 2 (as an example) that shows this information. Also, the authors should add another sub-plot if they want to talk about the total water consumption in figure 3. In the current format, readers have to calculate everything to understand the numbers.*

**Author response:** Table 2 indeed contains absolute numbers in terms of population totals, future discharges and water consumption at the sub-basin level for three timesteps. In the text related to Figure 3 and Table 2, we discuss the changes between these timesteps in terms of growth percentages, as it is the relative changes that allow us to compare between big and small sub-basins. This relative comparison explains why some sub-basins have similar trends, despite having vastly different absolute population and discharge totals. Nonetheless, we understand that this inconsistency in unit notation between Table 2 and the text can cause confusion. We will therefore add the relative change compared to the reference period (between parenthesis) behind the absolute numbers in Table 2. With regards to Figure 3, the total water consumption in each timestep and scenario can be understood from the figures by the top of the stacked bars. The total upper Indus water consumption is made up of the sum of all stacked sectoral water use, or the sum of all sub-basins. Moreover, the total consumption per sub-basin in absolute values can also be obtained from Table 2 'Water Consumption (km/3)'. We will clarify this in the caption of the Figure 3 and add additional references to the relevant data in Table 2.

*16.) Line 222: suddenly jump to Table 2 before providing complete information for Figure 3B.*

**Author response:** Table 2 provides complementary quantitative information on a multitude of factors (climate change, population change, consumption change). In this particular case, the population data of Table 2 is required to explain the patterns found in Figure 3B (namely, why several sub-basins show more growth than others). We therefore prefer to leave this as it is. We will however restructure the text so that the transition from Table 2 to Figure 3B is smoother.

*17.) The authors do not discuss figure 3A.*

**Author response:** We will add an interpretation of Figure 3A in line 255. It demonstrates the effect of urbanization on the spatial distribution of water demand.

*18.) Lines 233-234: "the relative growth ... the annual average", why does it happen? Is it only because of the agricultural sector?*

**Author response:** The growth in water consumption is largely driven by increasing water demands for the domestic and industrial sectors. In our representation, this demand is steady throughout the year. Agricultural water use, on the other hand, is high in the wet season, but very limited in the dry season. Thus, the relative change in total water consumption is highest in the season in which agricultural water use plays the smallest role (i.e. the dry season). We will expand the explanation on this process in line 233-234.

*19.) Lines 235-236: "Figure 3 shows ... both scenarios", is there any reason for it?*

**Author response:** Similar to the response to comment 17, domestic and industrial water use accounts for an increasing share of the total water consumption in the upper Indus, which do not vary throughout the year to the same degree as agricultural water demands. Hence, the differences in water demand between the seasons decrease.

*20.) Lines 238-239: "Table 2 ... 2060-2080 period" any physical interpretations?*

**Author response:** Higher temperatures due to climate change drive increased glacial melting in High Mountain Asia, thereby providing more meltwater and higher mountain water discharges. For a more detailed explanation of the biophysical processes behind this phenomenon we refer to Wijngaard et al. (2018). We will include this clarification in line 238-239.

*21.) Lines 252-257: Please exactly define whether Figure 4A or 4B is read.*

**Author response:** The first segment of this paragraph refers to figure 4A. The second segment refers to 4B. We will clarify this in the manuscript in the appropriate lines.

*22.) Lines 266-269: only three lines for the description of figure 5?*

**Author response:** We present Figure 5 to illustrate the effect of upstream consumption on (future) environmental flows. The few lines dedicated to this plot highlight our key message from this figure, namely that environmental flows are likely to be affected heavily, but only in two subbasins. Specific information on how environmental flows are defined and operationalized (and thus should be interpreted) is outlined in the methodology (line 172-178). We will expand the description in the text of the results section for this figure to highlight more patterns for other sub-basins as well. If the reviewer has specific suggestions, we would be happy to elaborate on any other points as takeaways from the figure.

*23.) Line 368: "Indus Water Treaty", this is the first time it appears in the manuscript! Provide more information about the treaty in the case study section. The study is on a transboundary river; however, this is the first paragraph that discusses the results from a transboundary view!*

**Author response:** We kindly refer to the response to comments 3, in which we describe several changes to our discussion and results section, including expanding the current reflection on the implications of our

study results for the Indus Water Treaty. Moreover, we refer to the response to comment 7, where we discuss the addition of a new section that describes present water management and treaties.

*24.) Line 377: some summary on the work, case study, and method should be provided here before writing about the findings.*

**Author response:** We will add a short summary on the research before stating the main conclusion.

*25.) As a transboundary study, the study needs to provide understanding and insights for the water management of each country in the basin.*

**Author response:** We kindly refer to the response to comment 3 where we outline the changes we will make in the results and discussion section to reflect on the consequences for the riparian states.

*26.) The caption of Figure 4: "Top" and "bottom" needs to change to "A" and "B".*

**Author response:** We thank the reviewer for this sharp observation and will address it accordingly in the manuscript.

*27.) The caption of Table 2: explain the "Mid" and "Late" in the caption.*

**Author response:** Mid and late refer to the Mid (2030-2050) and Late (2060-2080) time periods used in our study and explained at first occurrence in the methodology (see line 185-187). We will add this clarification to the caption of Table 2.

**References**

1.) Sivapalan, M., Savenije, H. H., & Blöschl, G. (2012). Socio-hydrology: A new science of people and water. Hydrol. Process, 26(8), 1270-1276.
2.) Munia, H., Guillaume, J. H. A., Mirumachi, N., Porkka, M., Wada, Y., & Kummu, M. (2016). Water stress in global transboundary river basins: significance of upstream water use on downstream stress. Environmental Research Letters, 11(1), 014002.
3.) Basharat, M. (2019). Water Management in the Indus Basin in Pakistan: Challenges and Opportunities. In Indus River Basin (pp. 375-388). Elsevier.
4.) Wescoat Jr, J. L., Siddiqi, A., & Muhammad, A. (2018). Socio-hydrology of channel flows in complex river basins: Rivers, canals, and distributaries in Punjab, Pakistan. Water Resources Research, 54(1), 464-479.
5.) Qamar, M. U., Azmat, M., & Claps, P. (2019). Pitfalls in transboundary Indus Water Treaty: a perspective to prevent unattended threats to the global security. npj Clean Water, 2(1), 1-9.
6.) Ali, S. S., & Bhargava, M. B. (2021). Hydro-diplomacy Towards Peace Ecology: The Case of the Indus Water Treaty Between India and Pakistan. In Decolonising Conflicts, Security, Peace, Gender, Environment and Development in the Anthropocene (pp. 591-613). Springer, Cham.

---

## Author Comment (AC2)

**Response to Reviewer 2**

We thank the reviewer for the concise and constructive comments. After careful consideration we provide a point-by-point reply to each of the comments in the text below. We are confident that the reviewers comments and corresponding suggested changes will improve the quality of the paper. The reviewer's original comments are numbered and in italics for clarity. References which do not appear yet in the main manuscript are listed at the end of this document We will modify the manuscript according to our responses.

1.) *The paper provides a comparative analysis of the impact of upper Indus water usage on downstream water availability under future climate change and socio-economic development. The analysis was done in sub-basin scale and under seasonal variability which is potentially interesting. Though the analysis is done on transboundary basin, there is no discussion how the results can add value on transboundary water management.*

**Author response:** The results of our study show several novel implications for future transboundary water management in the Indus basin. Foremost, previous quantitative studies on transboundary upstream-downstream linkages (Munia et al., 2016; Munia et al., 2017; Munia et al., 2020; Viviroli et al., 2020) evaluated the upper Indus as one aggregated sub-basin. These coarser assessments only provided insight into the development of such trends for the Indus basin at large. Instead, our regional assessment makes a disaggregated assessment for each major Indus tributary in the upper Indus and corresponding downstream areas. This allows our study to look at the trends of future upstream-downstream linkages within the Indus basin. Our approach is especially commendable for providing new quantitative insights into the development of upstream-downstream linkages for Indus sub-basins that are shared between multiple states. Our study thus looks for the first time at the scale of individual tributaries, where transboundary upstream-downstream tensions occur and should be addressed. This method allows for the identification of hotspots within the Indus basin where future transboundary upstream-downstream issues are likely to occur and ensuing hydropolitical tensions are likely to arise between basin countries.

Within the present discussion we highlight such sub-basin level transboundary implications specifically for the Kabul and Jhelum subbasins (largely administered by Afghanistan and India, respectively) and the increasingly populous downstream plains of the Pakistani Punjab (see line 353-370). In addition, we use this discussion to reflect on the future of the Indus Water Treaty. Nonetheless, upon revisiting our document, we acknowledge that the reflection on the larger transboundary implications of the patterns found in our study (line 342-370) can be made more explicit.

To highlight the value added by our novel approach to transboundary water management in the Indus and beyond, we suggest the following changes to the manuscript:

- First of all, we will expand the discussion section dedicated to the new insight provided by our disaggregated approach versus previous lumped approach (line 342-352). We will do so by linking this segment explicitly to new insights gained in our study on transboundary upstream-downstream linkages, as compared to insights provided by previous studies at coarser resolution.
- Second, both the Jhelum and Kabul sub-basins are shared between multiple riparian states. The consequences of changing upstream water use in the upper Indus and the emergence of impact hotspots in the lower Indus (discussed in line 342-360) are thus transboundary in character. However, these changing upstream-downstream linkages are currently described only in geographical terms and an explicit link to the riparian states involved is not made in this segment of the discussion. We will therefore add context on the relevant riparian states here, similar to the way in which we did in line 361-370.

- Third, we will expand the discussion on the consequences that changing upstream consumption and downstream water availability will have for water management and water sharing treaties between the riparian states (line 367-370), such as the Indus Water Treaty. This reflection will be aided by the inclusion of an overview of present water management practices and treaties in the 'Methods and Materials' section (see the response to comment 3).
- Lastly, we think that the tail section of our discussion and conclusion largely reflects on lessons learnt for adaptation towards water and food SDGs and for future hydrological modelling (line 371-375 and 393-402). We will revise these sections to also provide key highlights for transboundary water management. These highlights will be based on the discussion in lines 342-370, expanded with the changes mentioned above.

*2.) I found the method and material section complex and difficult to follow. May be using separate sections for data and scenarios will help.*

**Author response:** This point was also raised by Reviewer #1. The present 'Materials & Methods' section contains a separate section for the scenarios, while most data is introduction within the 'Analysis and data sources' segment. However, the scenarios that we use in our study are quantitative and spatially explicit and this has caused some overlap to occur between the 'narrative' side of the scenario and 'forcing data' side of the scenarios. Upon revisiting the document we do understand that the current blended introduction of scenario narratives and forcing data makes it more difficult to understand our methodology.

In light of both reviewer comments, we suggest the following changes:

- Primarily, we will restructure the 'Methods & Materials' section. We will split the 'Spatially explicit scenario context' section (line 84-100) to only contain the qualitative, narrative-related information of the scenarios. The forcing data information of the scenarios will be moved to a new sub-section '2.3.1. Scenario forcing data' at the start of the 'Analysis and data sources' section. Here we will explain the input/forcing data that was used to generate the hydrological and water consumption data used directly in our water accounting assessment.
- Additionally, we will add references to Table 1 within the 'Analysis and data sources' section to better clarify what data was exactly used in each part of the analysis. We will also adapt the terminology in this table to be consistent with that used in the text.
- Lastly, we will revise the general overview of the overall approach of our methodology (lines 78-83) to more clearly outline the division between scenario, data and analysis.

*3.) In the discussion section the current water management needs to be discussed. Moreover, the novelty of this analysis needs to discuss clearly.*

**Author response:** We agree that in our manuscript the discussion on present day water management practices, especially from a transboundary perspective, is limited. A similar concern was raised by reviewer #1, who suggested adding a 'case study' sub-section in the 'Methods and Materials' section to provide an overview of current transboundary water management practices and the role of water sharing accords, such as the Indus Water Treaty. We will implement this suggestion and use this overview to also expand the current reflection (see line 367-370) on the implications of our results for present and future water sharing treaties and water management in the Indus basin (also see response to comment 1).

*4.) Future of the upper Indus basin's water availability is highly uncertain in the long run, mainly due to the large spread in the future precipitation projections. Despite large uncertainties in the future climate and long-term water availability, basin-wide patterns and trends of seasonal shifts in water availability are consistent across climate change scenarios.*

**Author response:** We thank the reviewer for this important comment on the consideration of uncertainty in hydroclimatic projection. Previous studies have shown that rising temperatures under climate change will intensify glacial melt in this region and subsequently increase seasonal mountain water discharges for the century to come (Wijngaard et al., 2018; Lutz et al., 2014). Since all climate change projections used in this study assume an increase in regional temperatures (~2°C in RCP4.5 and ~5°C in RCP8.5, see Lutz et al., 2016), the surge in meltwater, and thus the increase in surface water availability across the Indus basin, is also consistent across all climate change scenarios and models, despite the large uncertainty found in regional precipitation projections. We will clarify this important sidenote within the corresponding segments of the result and discussion sections.

*5.) There is no mention of green water and its importance, nor is the capacity of green water to partially substitute for blue water needs ignored. At least in the discussion section this limitation must be discussed, and the possible implications for the findings.*

**Author response:** The SPHY model from previous research was used to simulate the hydrology of upper Indus sub-basins from a biophysical point-of-view (Wijngaard et al., 2017). The subsequent discharge projections hence do not consider societal water withdrawals/consumption in the upstream areas. Instead, these anthropogenic factors, including the blue water consumption from agriculture, were determined independently using other simulation models and combined with the naturalized SPHY hydrology in our water accounting framework. The LPJmL model that we used to simulate the blue water footprint (i.e. consumptive water use for irrigation) will only apply irrigation water if the crop water requirements cannot be met with the water available through precipitation (Bondeau et al., 2007). Hence, the blue water footprint used in our study is already implicitly corrected for green water availability. However, the SPHY model does simulate a water balance at the grid cell level, in which evapotranspiration from a natural vegetation layer is accounted for (see Terink et al., 2015). A green water footprint is therefore already subtracted from the discharge projections of upper Indus sub-basins provided by the SPHY model. To avoid double accounting for green water, we only take the blue water needs from LPJmL. We acknowledge that in our present document this implicit role of green water is not clear. We will therefore add an explanation to clarify the processes described above in lines 121-124 and lines 144-162.

*6.) In this study Greenhouse emission and socioeconomic development scenarios are decoupled, which can introduce further uncertainty to the results, acknowledging that including SSPs is a strong point of this study. Maybe a discussion can help potential users.*

**Author response:** Our study considers both climatic and socio-economic drivers. We have therefore used the SSP and the RCP frameworks together. The coupling between the frameworks used in our study (i.e. SSP1-RCP4.5 and SSP3-RCP8.5) has been employed in earlier regional assessments (Wijngaard et al., 2018) and are among the most frequent combinations in global integrated RCP-SSP research (see O'Neill et al., 2020). However, as is stated in line 87-89, these couplings are extreme scenarios (best-case and worst-case) that constrain the bandwidth of plausible future development as indicated by state-of-the-art climate models and regionally downscaled socio-economic projections. Using our decoupled approach, more

moderate combinations between the RCPs and SSPs, or asymmetric scenarios (e.g. pessimistic socio-economic development with optimistic climatic development) are also possible at the regional level. Although such scenario choices will lead to different outcomes, it is likely that these would remain within the range of the results found in our current study, as these are based on scenarios that represent the boundaries of plausible future developments. We will acknowledge this uncertainty in the limitations section of the discussion (line 330-342) and reflect on its implications.

*7.) In Table 2 results are presented in number, while in the text it is in percentage. It will be better to adopt one method to avoid confusion.*

**Author response:** This point was also raised by reviewer #1. Table 2 indeed contains absolute numbers in terms of population totals, future discharges and water consumption at the sub-basin level for three timesteps. In the text related to Table 2, we discuss the changes between these timesteps in terms of growth percentages, as it is the relative changes that allow us to compare between big and small sub-basins. In the end, this relative comparison is valuable to explain why some sub-basins have similar trends in Figure 3, despite the vast differences between them in terms of total population and discharge. Nonetheless, we do understand that this inconsistency in unit notation between Table 2 and the text can cause confusion. We will therefore add the relative change compared to the reference period (between parenthesis) after the absolute numbers in Table 2.

*8.) In the discussion section a separate section on limitation of the analysis will be benefited.*

**Author response:** The current version of the discussion first discusses the limitations of our approach and subsequent opportunities for future research (line 310-342) and then discusses the implications of the work (line 343-375). We will make this distinction clearer by adding sub-section headers.

**References**

1.) Terink, W., Lutz, A. F., Simons, G. W. H., Immerzeel, W. W., & Droogers, P. (2015). SPHY v2. 0: Spatial processes in Hydrology. Geoscientific Model Development, 8(7), 2009-2034.
2.) O'Neill, B. C., Carter, T. R., Ebi, K., Harrison, P. A., Kemp-Benedict, E., Kok, K., ... & Pichs-Madruga, R. (2020). Achievements and needs for the climate change scenario framework. Nature climate change, 10(12), 1074-1084.